# Enhance the Visual Representation via Discrete Adversarial Training

**Xiaofeng Mao**[†]   **Yuefeng Chen**[†]   **Ranjie Duan**[†]   **Yao Zhu**[‡*]   **Gege Qi**[†]
**Shaokai Ye**[§]   **Xiaodan Li**[†]   **Rong Zhang**[†]   **Hui Xue**[†]
[†]Alibaba Group, [‡]Zhejiang University, [§]EPFL
{mxf164419,yuefeng.chenyf,ranjie.drj}@alibaba-inc.com

## Abstract

Adversarial Training (AT), which is commonly accepted as one of the most effective approaches defending against adversarial examples, can largely harm the standard performance, thus has limited usefulness on industrial-scale production and applications. Surprisingly, this phenomenon is totally opposite in Natural Language Processing (NLP) task, where AT can even benefit for generalization. We notice the merit of AT in NLP tasks could derive from the discrete and symbolic input space. For borrowing the advantage from NLP-style AT, we propose Discrete Adversarial Training (DAT). DAT leverages VQGAN to reform the image data to discrete text-like inputs, i.e. visual words. Then it minimizes the maximal risk on such discrete images with symbolic adversarial perturbations. We further give an explanation from the perspective of distribution to demonstrate the effectiveness of DAT. As a plug-and-play technique for enhancing the visual representation, DAT achieves significant improvement on multiple tasks including image classification, object detection and self-supervised learning. Especially, the model pre-trained with Masked Auto-Encoding (MAE) and fine-tuned by our DAT without extra data can get **31.40** mCE on ImageNet-C and **32.77%** top-1 accuracy on Stylized-ImageNet, building the new state-of-the-art. The code will be available at https://github.com/alibaba/easyrobust.

## 1  Introduction

Nowadays, Deep Neural Networks (DNNs) has achieved excellent performance surpassing humans in most computer vision tasks. Although remarkable progress has been made, the success of DNNs is actually a false sense when i.i.d hypothesis is not satisfied in wild. Researchers have shown that deep models fail in most circumstances including adversarial perturbations [1], common corruptions [2], colors or textures changing [3, 4], etc. There is still a long way to make DNNs closer to the robust human perception.

A possible way towards robust machine perception can be Adversarial Training (AT) [5], which automatically finds failure input cases of DNNs and augment online with these cases for fixing "bugs". With online augmentation of adversarial examples, AT greatly enhances the adversarial robustness, and helps for learning perceptually-aligned representations [6] with good interpretability [7, 8] and transferability [9]. However, AT is double-edged, which meanwhile degrades the standard performance caused by problematic regularization [10]. Such problematic regularization makes the decision boundaries over-smoothed and enlarges indecisive regions.

Surprisingly, previous works [11, 12] observe a strange phenomenon that AT behaves conversely in Natural Language Processing (NLP) tasks. By automatically finding adversarial textual inputs,

---

*Corresponding author: Yao Zhu (E-mail: ee_zhuy@zju.edu.cn)

36th Conference on Neural Information Processing Systems (NeurIPS 2022).

AT will not hurt the accuracy and even benefit for both generalization and robustness of language models. This phenomenon motivates us considering whether the merit of NLP-style AT can be transferred to vision tasks. We notice such merit could derive from the unique data organizing form of language models. To be specific, an adversarial image perturbed in continuous pixel space actually differs with the truly "hard" examples appeared in real world. Contrarily, text space is discrete and symbolic, where adversarial text is practically existing when a typo is made by humans. Learning on such adversarial text will obviously improve the generalization on other more texts with confusing typos. Therefore, we borrow the symbolic nature of languages, and apply it on CV tasks by discretizing continuous images into a more meaningful symbolic space. Afterwards, AT is conducted for minimizing the maximal risk on such text-like inputs with symbolic adversarial perturbations.

In this paper, we propose Discrete Adversarial Training (DAT), a new type of adversarial training which aims to improve both robustness and generalization of vision models. DAT leverages VQ-GAN [13] to learn a vocabulary of visual words, also known as image codebook. For a continuous image input, each encoded patch embedding is replaced with its closest visual word in the codebook, and represented as a corresponding index. Then the image is transformed to a sequence of symbolic indices similar with language input. For generating adversarial examples based on such symbolic sequence, direct use of optimization methods in NLP like combinatorial optimization [14, 15, 16] or synonym substitutions [14, 16] can be challenging. The reason lies in: 1) the large search space of images and 2) the non-existence of synonym in visual codebook. To make it more efficient, DAT adopts a gradient-based method which assumes the backward adversarial gradient goes straight-through the complex discretization process, thus gradients on discretized image can be copied to original input. Then it use one-step search along the direction of estimated gradient such that the discrete representation will altered adversarially during the discretization process, resulting a discrete adversarial examples. Finally the discrete adversarial examples is fed into models for training. Different from AT which always adds $l_p$ bound on augmented adversarial examples, DAT affects the discretization process to produce diverse adversarial inputs beyond $l_p$ bound for training. We show in ablation experiment that DAT not only enhances the robustness on $l_p$ bounded attacks, but also is partly beneficial in defense of unrestricted semantic attacks [17, 18]. The overall pipeline of discrete adversarial training is shown in Figure 1.

We further give an analysis to explain the effectiveness of our DAT from the perspective of distribution. By comparing the distributional difference of training examples in AT and DAT with clean images, we find the discrete adversarial examples in our DAT are much closer to the clean distribution. Such ability of generating "in-distribution" adversarial examples makes DAT can improve the visual representation learning on multiple vision models and tasks, with no sacrificing of clean accuracy.

Our contributions are summarized below:

- To the best of our knowledge, we appear to be the first to transfer the merit of NLP-style adversarial training to vision models, for improving robustness and generalization simultaneously.

- We propose Discrete Adversarial Training (DAT), where images are presented as discrete visual words, and the model is training on example which has the adversarially altered discrete visual representation.

- DAT achieves significant improvement on multiple tasks including image classification, object detection and self-supervised learning. Especially, it establishes a new state-of-the-art for robust image classification. By combining MAE [19] pre-training and DAT fine-tuning, our ViT-Huge [20] achieves 31.40 mCE on ImageNet-C [2] and 32.77% top-1 accuracy on Stylized-ImageNet [3].

## 2   Related Work

**Adversarial Training**   Adversarial Training (AT)   [5] is first proposed to improve robustness by training models with adversarial examples. As one of the most effective defense, existing works [21, 22] have suggested a trade-off between adversarial and clean accuracy in AT. Despite great efforts [23, 24, 25] have been made for mitigating this trade-off, the bad generalization of AT still cannot be fully remedied till now. In opposite way, some other works [26, 27] use AT to improve the clean accuracy rather than adversarial robustness. The most close work to ours is the AdvProp [26], which splits batch norms to prevent the mixed statistics of clean and adversarial

examples, thus learns better adversarial feature for generalization. Pyramid AT [28] makes AT specific to ViTs by crafting pyramid perturbation with Dropout enabling, yielding imporved performance. However, these methods are only applicable under specific models or tasks. VILLA [29] is also a representation enhancement technique using AT. But contrary to us, it applies vision-style AT only for vision-and-language representation learning. AGAT [30] is another kind of AT beyond pixel space, which perturbs images along attributes, however it has strict requirement of attributes annotation.

**Adversarial Augmentation**   By borrowing the idea of AT, some previous works [31, 32, 33] propose to search augmentations adversarially for improving the hardness of training examples. Adversarial AutoAugment [31] uses augmentation policy network to produce hard augmentation policies on a pre-defined policy space. AugMax [32] mixes multiple randomly sampled augmentation operators like AugMix [34], by using adversarially learned mixing factors. AdA [33] optimizes the parameters of image-to-image models to generate adversarially corrupted augmented images. MaxUp [35] uses the worst augmented data of each data point in a set of random perturbations or transforms for training. However, these methods create adversarial inputs indirectly and rely on pre-defined augmentations or translation models. They cannot perturb the images locally. Instead, our DAT directly modifies the image with no need of pre-defined transforms, and can craft local perturbations on images, which is more elaborate.

**Discrete Visual Representation Learning**   Early technique [36, 37] of Bag-Of-Visual-Words (BOVW) model has shown great power of discrete representation in visual understanding. VQ-VAE [38] uses DNNs to learn neural discrete representations, also known as visual codebook, by generative modeling the image distribution. Recently, the idea of discrete representations learning has been widely emerged in many vision tasks. In most Masked Image Modeling (MIM) methods [39, 40], visual codebook is needed for BERT-like self-supervised pretraining. For image classification, discrete representations strengthen the robustness by preserving the global structure of an object and ignoring local details [41]. For image synthesis, adversarial and perceptual objective can be added to VQ-VAE for learning perceptually-aligned visual codebook [13]. With growing power of generative models, now a well trained VQGAN can produce vivid images with 0.58 FID [42]. In this work, we use VQGAN for the discretization process in our DAT. As a powerful generative model, VQGAN can greatly reduce the information loss in reconstruction process and ensure the high-quality of the generated discrete adversarial examples.

## 3   Method

### 3.1   Traditional Adversarial Training

We take the typical image classification task as an example to show the formulation of Adversarial Training (AT). Consider the training image and label set $\mathcal{D} = \{x_i, y_i\}_{i=1}^{n}$, and a classifier $F$ with learnable parameters $\theta$, the classification objective is always a cross-entropy loss $\mathcal{L}(x, y, \theta)$. Adversarial Training (AT) finds the optimal $\theta$ by solving a minimax optimization problem:

$$\min_{\theta} \mathbb{E}_{(x,y) \sim \mathcal{D}} \left[ \max_{\delta} \mathcal{L}(x + \delta, y, \theta) \right] \quad s.t. \|\delta\|_p < \epsilon, \tag{1}$$

where the inner optimization finds the perturbations $\delta$ on per-pixel values for maximizing the loss, and the outer minimization update $\theta$ to improve the worst-case performance of the network w.r.t. the perturbation. $\| \cdot \|_p$ constraints the $p$-norm of $\delta$ to a small value $\epsilon$. A problem is that AT finds the failure case i.e. $x + \delta$ in continuous pixel space. However, human does not create or recognize images from complex pixel values, but from discrete semantic concepts. Although adversarial examples can successfully fool the models, they are still different from the real "hard" examples appeared in practice.

### 3.2   Discrete Adversarial Training

#### 3.2.1   Image Discretization by Visual Codebook

For discrete adversarial training, it is desirable first to learn an expressive visual codebook and represent the training image set in discrete space. We utilize VQGAN [13] for image discretization. More

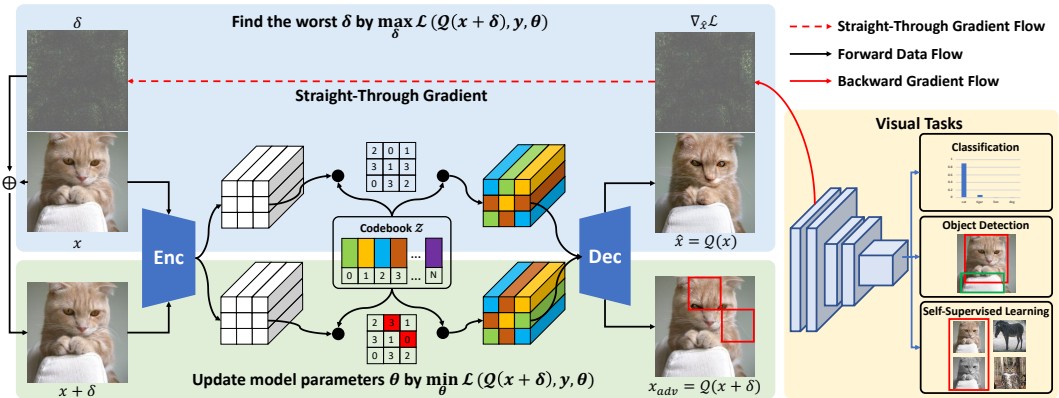

Figure 1: The overall pipeline of Discrete Adversarial Training (DAT).

precisely, consider a continuous image $x \in \mathbb{R}^{H \times W \times 3}$, VQGAN learns an encoder $\mathsf{Enc}_\phi (\cdot)$, decoder $\mathsf{Dec}_\psi (\cdot)$ and quantization $\mathsf{q}_{\mathcal{Z}} (\cdot)$. $\mathsf{Enc}_\phi$ is a convolutional model which maps $x$ to intermediate latent vectors $v = \mathsf{Enc}_\phi(x) \in \mathbb{R}^{(h \times w) \times d}$, where $h, w$ is the height, width of the intermediate feature map and $d$ is the latent dimension. Subsequently, $\mathsf{q}_{\mathcal{Z}}(\cdot)$ learn a codebook $\mathcal{Z} = \{z_k | z_k \in \mathbb{R}^d\}_{k=1}^K$, such that each latent vector $v_{ij} \in \mathbb{R}^d$ can be quantized onto its closest codebook entry $z_k$ as:

$$v_{\mathsf{q}} = \mathsf{q}_{\mathcal{Z}}(v) := \left( \underset{z_k \in \mathcal{Z}}{\arg \min} \, \|v_{ij} - z_k\| \right) \in \mathbb{R}^{h \times w \times d}, \tag{2}$$

where $i, j$ present each location in feature map. Then the decoder $\mathsf{Dec}_\psi (\cdot)$ outputs the reconstruction image $\hat{x}$ from the quantized vectors $v_{\mathsf{q}}$ by:

$$\hat{x} = \mathsf{Dec}_\psi (v_{\mathsf{q}}). \tag{3}$$

VQGAN is trained by minimizing the reconstructed difference between $\hat{x}$ and $x$. More details can be referred to Appendix A. So far, given a continuous image $x$, we can get its corresponding discrete reconstruction $\hat{x}$. For simplicity, we use $\mathcal{Q}$ to stand for the above image discretization process, and then we have $\hat{x} = \mathcal{Q}(x)$.

### 3.2.2 Discrete Adversarial Training

Based on the definition in Sec 3.2.1, we can generate discrete adversarial examples in inner maximization step of AT. By slightly modifying the Eq 1, the objective of DAT is formulated as:

$$\min_\theta \mathbb{E}_{(x,y) \sim \mathcal{D}} \left[ \max_\delta \mathcal{L}(\mathcal{Q}(x + \delta), y, \theta) \right], \tag{4}$$

where $\mathcal{Q}$ transforms the continuous pixel space to discrete input space. We delete the constraint term since there is no need to bound the per-pixel values of $\delta$. Suppose that $\mathcal{Q}$ is an ideal discretizer with no information loss in discretization process. The problem lies how to find the worst $\delta$ for maximizing the classification loss. Similar with traditional AT, we can use gradient-based methods to approximate $\delta$ by:

$$\delta \simeq \alpha \nabla_x \mathcal{L}(\mathcal{Q}(x), y, \theta), \tag{5}$$

where $\alpha$ determines the magnitude of the perturbations along the gradient direction. We set $\alpha = 0.1$ by default in DAT. To expand $\nabla_x \mathcal{L}(\mathcal{Q}(x), y, \theta)$ by chain rule, we have four partial derivative terms as follows:

$$\nabla_x \mathcal{L}(\mathcal{Q}(x), y, \theta) = \frac{\partial \mathcal{L}}{\partial \hat{x}} \cdot \frac{\partial \hat{x}}{\partial v_{\mathsf{q}}} \cdot \frac{\partial v_{\mathsf{q}}}{\partial v} \cdot \frac{\partial v}{\partial x} \tag{6}$$

Through analysing the feasibility of each term, we find only $\frac{\partial v_{\mathsf{q}}}{\partial v}$ is hard to solve as the non-differentiable nature of Eq 2. Fortunately, as proposed in previous work, a straight-through gradient estimator [43, 44] can be used by copying the gradients from $v_{\mathsf{q}}$ to $v$. By replacing $\frac{\partial \hat{x}}{\partial v_{\mathsf{q}}} \cdot \frac{\partial v_{\mathsf{q}}}{\partial v}$ with $\frac{\partial \hat{x}}{\partial v}$, we can simplify the Eq 6 to $\nabla_x \mathcal{L}(\mathcal{Q}(x), y, \theta) = \frac{\partial \mathcal{L}}{\partial \hat{x}} \cdot \frac{\partial \hat{x}}{\partial v} \cdot \frac{\partial v}{\partial x}$, which has derivative everywhere.

**Algorithm 1:** Pseudo code of DAT

---

**Input:** Classifier $F$; Pre-trained discretizer $\mathcal{Q}$; A sampled mini-batch of clean images $x$ with labels $y$; attack magnitude $\alpha$.

**Output:** Learned network parameter $\theta$ of $F$

1: Fix the network parameters of $\mathcal{Q}$
2: **for** each training steps **do**
3:     $\hat{x} \leftarrow \mathcal{Q}(x)$                                                     //Get the discrete reconstruction $\hat{x}$
4:     $\delta \leftarrow \alpha\nabla_{\hat{x}}\mathcal{L}(\hat{x}, y, \theta)$                       //Estimate the adversarial perturbations
5:     $x_{adv} \leftarrow \mathcal{Q}(x + \delta)$                     //Generate discrete adversarial examples
6:     Minimize the classification loss w.r.t. network parameter
       $\arg\min_\theta \mathcal{L}(x_{adv}, y, \theta)$
7: **end for**

---

Although the solution seems workable theoretically, the huge cost makes it impractical on large-scale vision tasks. The bottleneck mainly lies on that $\frac{\partial\hat{x}}{\partial v}$ and $\frac{\partial v}{\partial x}$ require the adversarial gradients backward through Enc and Dec. Actually, a generator capable of producing high-quality images always has a large amount of parameters. Compared with original adversarial training which only needs $F$ for gradient calculation, it requires more than tripled GPU memory and computation cost.

To solve this problem, we propose an efficient alternative solution. Since $\hat{x} \simeq x$ is empirically observed for an ideal discretizer $\mathcal{Q}$, we can also use a straight-through estimator between $\hat{x}$ and $x$, which is given by

$$\nabla_x\mathcal{L}(\mathcal{Q}(x), y, \theta) = \frac{\partial\mathcal{L}}{\partial\hat{x}} \cdot \frac{\partial\hat{x}}{\partial x} \simeq \frac{\partial\mathcal{L}}{\partial\hat{x}}. \tag{7}$$

Finally, we can solve the worst $\delta$ by $\nabla_{\hat{x}}\mathcal{L}(\hat{x}, y, \theta)$. By this way, the computation cost of DAT has been largely reduced. Compared with original adversarial training, it only has extra computation cost on VQGAN forward, which is relatively controllable.

For clarity, let us restate the pipeline of our DAT. For each training image $x$, DAT first use VQGAN to get discrete reconstruction $\hat{x}$. By feeding $\hat{x}$ to classifier $F$, a worst-case perturbation $\delta$ can be estimated by computing the gradient of $\hat{x}$ towards maximizing the classification loss. The perturbed image thus can be created by adding $\delta$ on original $x$. Finally, $x + \delta$ is discretized by VQGAN again and acts as the adversarial input, on which $F$ is trained by minimizing the classificaiton loss. The details of our DAT is summarized in Algorithm 1.

**Explaining the Effectiveness of DAT from the Perspective of Distribution** We give an empirical analysis to explain why DAT can improve the robustness and generalization without sacrificing clean accuracy. Previous work [26] has pointed out that the underlying distributions of adversarial examples are different from clean images. Training on both clean and adversarial images will force the Batch Normalization (BN) [45] to estimate an inaccurate mixture statistics of feature distribution, and thus impact the standard performance. We study this effect by sampling 1000 mini-batches in ImageNet validation set, and generate corresponding adversarial images for AT and our DAT. For each batch we calculate the mean and variance statis-

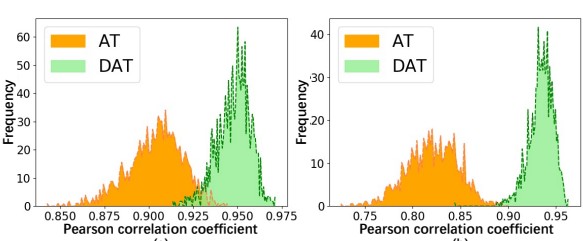

Figure 2: The frequency histogram of the Pearson correlation coefficient (PCC) between BN statistics on clean and adversarial images. Larger PCC value means smaller distributional difference with clean images. (a), (b) present the difference on mean and variance statistics respectively.

tics of last BN of ResNet50. Then the Pearson correlation coefficient (PCC) between the statistics on clean and adversarial examples is computed for showing the distributional difference. Figure 2 shows the frequency histogram of the distributional difference on 1000 mini-batches. For training samples of DAT, the peak of the histogram is at 0.95, which is greater than AT. It suggests that DAT

| Methods | ImageNet | Adversarial Robustness | | Out of Distribution Robustness | | | | | |
| | | FGSM | DamageNet | A | C↓ | V2 | R | Sketch | Stylized |
|---|---|---|---|---|---|---|---|---|---|
| ResNet50 [46] | 76.13 | 12.19 | 5.94 | 0.0 | 76.70 | 63.20 | 36.17 | 24.09 | 7.38 |
| + DAT (Ours) | **76.52** | **30.66** | **14.42** | **4.38** | **74.16** | **65.02** | **41.90** | **27.27** | **10.8** |
| DeepAugment [47] | 76.66 | 21.61 | 11.94 | 3.46 | 60.37 | **65.24** | 42.17 | 29.50 | 14.68 |
| + Augmix [34] | 75.82 | 27.05 | 19.60 | 3.86 | 53.55 | 63.63 | 46.77 | 32.62 | 21.23 |
| + DAT (Ours) | **77.10** | **35.32** | **22.86** | **6.86** | **50.82** | 65.14 | **47.88** | **34.98** | **21.89** |
| ViT [20] | 72.00 | 23.30 | 28.99 | 6.44 | 77.61 | 57.34 | 25.69 | 15.56 | 5.82 |
| DrViT [41] | 79.48 | 45.76 | 44.91 | 17.20 | 46.22 | 68.05 | 44.77 | 34.59 | 19.38 |
| AugReg-ViT [48] | 79.91 | 44.32 | 45.24 | 19.03 | 54.50 | 67.90 | 39.46 | 29.16 | 16.62 |
| + DAT (Ours) | **81.46** | **51.82** | **45.70** | **30.15** | **44.65** | **70.83** | **47.34** | **34.77** | **23.13** |
| MAE-H [19] | 86.90 | 60.16 | 64.36 | 68.18 | 33.92 | 78.47 | 64.12 | 49.08 | 26.36 |
| + DAT (Ours) | **87.02** | **63.77** | **70.42** | **68.92** | **31.40** | **78.82** | **65.61** | **50.03** | **32.77** |

Table 1: The results of DAT on image classification. Bold number indicates the better performance.

| Training Strategies | ImageNet | Adversarial Robustness | | Out of Distribution Robustness | | | | | |
| | | FGSM | DamageNet | A | C↓ | V2 | R | Sketch | Stylized |
|---|---|---|---|---|---|---|---|---|---|
| Normal [48] | 79.91 | 44.32 | 45.24 | 19.03 | 54.50 | 67.90 | 39.46 | 29.16 | 16.62 |
| Advprop [26] | 79.54 | **72.38** | 45.48 | 18.53 | 51.46 | 68.74 | 43.51 | 31.68 | 19.24 |
| Fast Advprop [27] | 79.02 | 70.52 | 44.87 | 17.86 | 53.31 | 67.09 | 41.84 | 29.42 | 18.39 |
| Pyramid AT [28] | **81.68** | 50.36 | 45.53 | 23.18 | 44.95 | 70.32 | 47.30 | **36.87** | 20.02 |
| Debiased [49] | 79.33 | 46.85 | 44.99 | 18.32 | 49.82 | 67.55 | 40.32 | 29.43 | 22.37 |
| **DAT (Ours)** | 81.46 | 51.82 | **45.70** | 30.15 | 44.65 | 70.83 | 47.34 | 34.77 | **23.13** |

Table 2: Comparison of DAT with other training strategies. We use AugReg-ViT as the base model.

generates discrete adversarial examples much closer with the clean distributions. Therefore, training on these examples will reduce the shift of clean distribution in AT, yielding both the robustness and generalization improvement.

## 4 Experiments

To demonstrate the versatility of our method, we experiment Discrete Adversarial Training (DAT) on multiple tasks including image classification, object detection and self-supervised learning.

### 4.1 Image Classification

**Implementation** We implement DAT on two representative architectures: ResNet50 [46] and ViTs [20]. For ResNet50, we first experiment DAT with vanilla training recipes using "robustness" library [2]. Then we combine DAT with other orthogonal robust training techniques: DeepAugment [47] and AugMix [34]. Only cross entropy loss is used for generating discrete adversarial examples. The JSD loss in AugMix is optimized merely on clean samples. For ViTs, we adopt ViT-B/16 as baseline models, which is trained by the recipes in AugReg [48]. Besides, we use DAT to conduct supervised finetuning on downstream ImageNet classification task based on a self-supervised ViT-Huge pretrained by MAE [19]. By default, we refer ViT to ViT-B/16 in all tables and figures.

**Benchmarks** The trained model is evaluated in three aspects: 1) in-distribution performance on ImageNet-Validation set; 2) adversarial robustness on white-box FGSM [50] and transfer-based black-box attack dataset DamageNet [51]; 3) out-of-distribution robustness on ImageNet(IN)-A, IN-C, IN-V2, IN-R, IN-Sketch and Stylized IN. Each of them represents a type of out-of-distribution scenario where the classifier is prone to make mistakes. IN-A [52] places the ImageNet objects in hard contexts; IN-C [53] applies a series of noise, blur, digital and weather corruptions; IN-R [47] collects online images with artificial creation, e.g., cartoons, graphics, video game renditions, etc; IN-Sketch [54] contains images described by sketches; Stylized IN [3] destroys the texture but maintains the shape feature by conducting style transfer on ImageNet images. Except for IN-C which is measured by mCE, we report the top-1 accuracy on all above datasets.

**Results** We report all results in Table 1. For fair comparison, we add DAT on base methods without modifying the original training hyper-parameters, such that improvement is entirely attributed to the

---

[2]https://github.com/MadryLab/robustness

| Method | ImageNet | Flowers | Linear Evaluation | | | | VOC Object Detection | | | ADE20K | |
| | | | CIFAR10 | Caltech101 | Cars | DTD | mAP | AP50 | AP75 | mIoU | Acc. |
|---|---|---|---|---|---|---|---|---|---|---|---|
| MoCov3 | 68.63 | 91.54 | 93.40 | 90.38 | 49.01 | 73.03 | 50.42 | 80.53 | 53.93 | 0.3508 | 75.64 |
| **+ DAT** | **69.60** | **93.15** | **95.16** | **91.42** | **53.09** | **73.55** | **51.92** | **80.97** | **56.04** | **0.3585** | **76.33** |
| SimCLR | 64.89 | 89.28 | 88.47 | 83.20 | 38.84 | **73.14** | 48.50 | 78.75 | 51.35 | 0.3396 | **75.61** |
| **+ DAT** | **65.47** | **90.14** | **89.97** | **85.09** | **39.42** | 72.93 | **48.83** | **79.27** | **51.95** | **0.3412** | 75.46 |
| SimSiam | 68.16 | **87.67** | 89.45 | 85.94 | 34.95 | 71.70 | 48.92 | 77.22 | 52.69 | **0.2212** | 68.57 |
| **+ DAT** | **68.41** | 86.93 | **91.70** | **87.03** | **35.10** | **73.03** | **51.73** | **79.69** | **55.74** | 0.2203 | **68.92** |

Table 3: The results of DAT on self-supervised learning.

DAT. For ResNet50, DAT achieves significant improvement on both clean accuracy, adversarial and out-of-distribution robustness. The improvement seems greater when combining DAT with DeepAugment and AugMix. For ViTs, we find DAT is compatible with other complex augmentations such as MixUp [55], CutMix [56] or RandAugment [57] used in AugReg, yielding greater improvement. Compared with plain ViT, AugReg-ViT and ViT with discrete representation called DrViT [41], DAT with AugReg achieves better performance. The best result is from ViT-Huge pretrained by MAE [19] and finetuned by DAT, which suggests DAT is also effective in downstream fine-tuning tasks.

Additionally, we also compare DAT with other robust training strategies in Table 2. AugReg-ViT is adopted as the baseline model. Most strategies, e.g., AdvProp [26] and Debiased [49] are proposed for ResNet with auxiliary BatchNorm. We show these methods cannot work properly on ViTs with only LayerNorm. Compared with Pyramid AT [28], our DAT has lower clean accuracy but yields stronger robustness. More results of strategies comparison on ResNet50 refer to Appendix B.1.

## 4.2 Self-Supervised Learning

**Implementation** We experiment DAT on three self-supervised methods: SimCLR [58], SimSiam [59] and recently proposed MoCov3 [60]. The discrete adversarial training is only conducted during pre-training stage, and the learned representation is evaluated on downstream tasks by standard pipeline [61]. We craft adversarial examples based on RoCL [62], which attacks the pre-training objective by maximizing the contrastive loss. For preventing the cost explosion, we pre-train 200 epochs for SimCLR, and 100 epochs for both SimSiam and MoCov3. The batch size used for SimCLR, SimSiam, MoCov3 is set as 1024, 512, 2048 respectively.

**Benchmarks** For self-supervised learning, we adopt ImageNet-1K for both training and in-distribution testing. Beyond that, in order to give more comprehensive assessment, we build three downstream tasks to metric the transferability of the learned visual representation. Specifically, the linear evaluation reports the top-1 accuracy on five classification datasets: Flowers, CIFAR10, Caltech101, Cars and DTD. For downstream evaluation of object detection, we present mAP, AP50, AP75 on Pascal VOC2007 [63]. ADE20K [64] is used for semantic segmentation task, and both the mean intersection over union (mIoU) and accuracy are reported.

**Results** As shown in Table 3, DAT can enhance the learned representation on ImageNet and get 0.97%, 0.58% and 0.25% improvement on MoCov3, SimCLR, SimSiam respectively. For excluding that DAT is not just over-fitting on ImageNet, we transfer the representations on downstream recognition, object detection, semantic segmentation tasks. The results suggest that DAT enhances the self-supervised representations with better transferability. In particular, MoCov3 with DAT achieves significant improvement in all downstream tasks. This conclusion echoed with previous works [9], which find adversarially robust models often perform better on transfer learning.

| Models | Training Strategy | COCO mAP | COCO-C mAP | Relative rPC (%) |
|---|---|---|---|---|
| EffDet-Lite0-320 [65] | Normal | 26.41 | 16.11 | 61.00 |
| | Det-Advprop | 26.34 | 16.38 | 62.19 |
| | **DAT (Ours)** | **27.32** | **17.89** | **65.48** |
| EffDet-Lite1-384 [65] | Normal | 31.50 | 19.43 | 61.68 |
| | Det-Advprop | 31.82 | 20.21 | 63.51 |
| | **DAT (Ours)** | **32.31** | **21.32** | **65.99** |
| YOLOv3-320 [66] | Normal | 35.91 | 18.39 | 51.21 |
| | Det-Advprop | 35.73 | 19.34 | 54.13 |
| | **DAT (Ours)** | **36.02** | **20.55** | **57.05** |
| YOLOv3-416 [66] | Normal | 40.30 | 21.19 | 52.58 |
| | Det-Advprop | **40.69** | 22.55 | 55.42 |
| | **DAT (Ours)** | 40.41 | **23.38** | **57.86** |

Table 4: The results of DAT on object detection.

| Models | Datasets | $f$ | $K$ | $d$ | FID | ImageNet | FGSM | A | C↓ | R | Stylized |
|---|---|---|---|---|---|---|---|---|---|---|---|
| VQGAN [42] | OpenImages [70] | 8 | 16384 | 4 | 1.14 | 76.52 | 30.66 | 4.38 | 74.16 | 41.90 | 10.80 |
| VQGAN [42] | OpenImages [70] | 16 | 16384 | 8 | 5.15 | 75.65 | 30.48 | 3.17 | 74.71 | 40.64 | 10.76 |
| VQGAN [42] | OpenImages [70] | 8 | 256 | 4 | 1.49 | 76.52 | 30.46 | 3.11 | 74.06 | 40.97 | 10.01 |
| VQGAN [13] | ImageNet [71] | 16 | 16384 | 256 | 4.98 | 75.88 | 30.28 | 2.65 | 75.38 | 40.66 | 9.58 |
| DALL-E [72] | Private | 8 | 8192 | 128 | 32.01 | 75.23 | 30.17 | 3.97 | 74.26 | 40.13 | 10.65 |

Table 5: Results of DAT based on different pretrained discretizer $\mathcal{Q}$. $f$ presents the downsampling factors. We use ResNet50 as base model and the subset of benchmarks in Sec 4.1 for evaluation.

## 4.3 Object Detection

**Implementation** We implement DAT on two popular detectors: EfficientDet [65] and YOLOv3 [66]. Object detection generally has two sub-tasks: classification and localization. Det-AdvProp [67] proposes to select more vulnerable sub-task to generate adversarial images, and use auxiliary BN for training. Instead, DAT regards the two sub-tasks as a whole, and attacks the overall detection loss to produce adversarial images. Besides, DAT does not modify BN in detectors. As the memory and computation cost of VQGAN increased quadratic with the input resolution, DAT become unaffordable when input size is large. Therefore, we only experiment DAT on lightweight version of EfficientDet and YOLOv3 with input size smaller than 512.

**Benchmarks** We train the detectors using the COCO 2017 object detection dataset [68] and evaluate them on COCO's validation set. COCO-C [69] is used for testing the robustness to natural corruptions. We report mAP on COCO and COCO-C as the clean and robust performance respectively. rPC [69] is used to to measure relative performance degradation under corruption.

**Results** We compare the detectors with DAT, Det-AdvProp [67] and vanilla training in Table 4. Although Det-AdvProp has been shown effective on EfficientDet with size larger than D0, we find on smaller detectors, the promotion of Det-AdvProp is subtle. On EfficientDet-Lite0 and YOLOv3-320, it even gets worse on COCO mAP than vanilla training. While our DAT achieves better result on both clean and corrupted input. EfficientDet-Lite0 with DAT get 27.3 and 17.89 mAP on COCO and COCO-C, resulting in 65.53% Relative rPC. By comparison, vanilla YOLOv3 models have lower Relative rPC, showing it is more vulnerable than EfficientDet at same clean mAP. After equipped with DAT, YOLOv3-320 can get 20.55 mAP on COCO-C, leading to 7.75% improvement on rPC.

## 4.4 Ablations

**Impact of the Codebook in DAT** A general understanding is that the codebook with larger $K$ should have stronger representation power. As suggested in Table 5, when the $K$ is reduced from 16384 to 256, the FID of generated images increases 0.35. We show DAT on larger codebook size can achieve better generalization. However, the clean accuracy seems no improvement with increasing of $K$. We think the reason is that the reconstruction quality has already met the demand of DAT. And the improvement on clean accuracy is saturated when $K$ further increases.

**Different Types of Discretizer $\mathcal{Q}$** Our work is based on the hypothesis of an ideal $\mathcal{Q}$ with no loss in discretization process. However, such perfect $\mathcal{Q}$ is not existing in practice. So we study if and how the ability of $\mathcal{Q}$ affects the performance of our DAT in Table 5. The results show DAT performs better by using discretizer with higher FID. We find clean accuracy declines 1.3% on DALL-E, showing it is important for choosing a powerful discretizer. Types of pre-training datasets have little impact. We find the discretizer pre-trained on OpenImages [70] and transferred to discretize ImageNet images, can also achieve comparable results.

**The Performance on Different Magnitude $\alpha$** We present the DAT results on ResNet50 and ViT with different magnitude $\alpha$ in Figure 3. $\alpha = 0$ means the model is only trained on images augmented by VQGAN with no adversarial training process. We show DAT with $\alpha = 0$ makes the models have high clean accuracy but the lowest robustness. With the increase of $\alpha$, there is still a robustness and accuracy trade-off. DAT with $\alpha = 0.4$ on ResNet50 achieves the best adversarial robustness but generalization is getting worse. By contrast, we surprisingly find ViT has the lower sensibility on $\alpha$. Even if ViT is trained on DAT with $\alpha = 0.4$, there is still no great drop of clean and OOD accuracy.

We suspect that ViT is more suitable for training on discrete examples, and the strong modeling ability makes ViT greater than ResNet50.

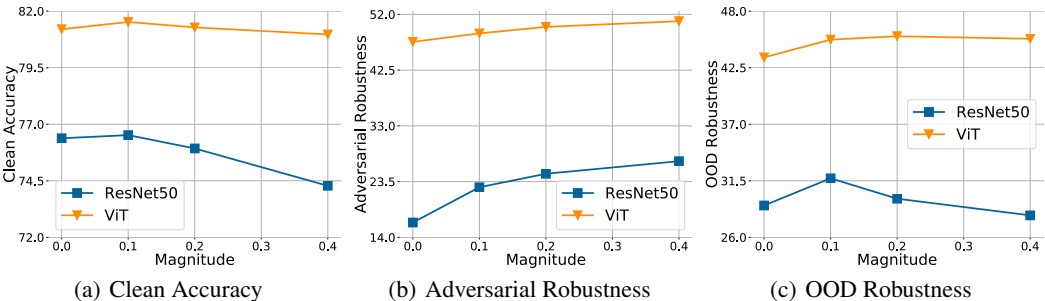

(a) Clean Accuracy  (b) Adversarial Robustness  (c) OOD Robustness

Figure 3: The performance of DAT with different magnitude $\alpha$ in Eq 5.

**Results on Stronger Attacks** In the main experiment, only one-step FGSM is used for examining the robustness under white-box adversarial attacks. To give more comprehensive evaluation, we additionally test our DAT under stronger AutoAttack [73], and two unrestricted attacks named Adv-Drop [17] and PerC-Adversarial [18]. The result is shown in Figure 4. DAT brings the improvement of robustness under all three attackers. ViT trained with DAT achieved extremely high robust accuracy on PerC-Adversarial, which suggests DAT also effects well in defense against unrestricted adversarial attacks. More details can be referred to Appendix C.

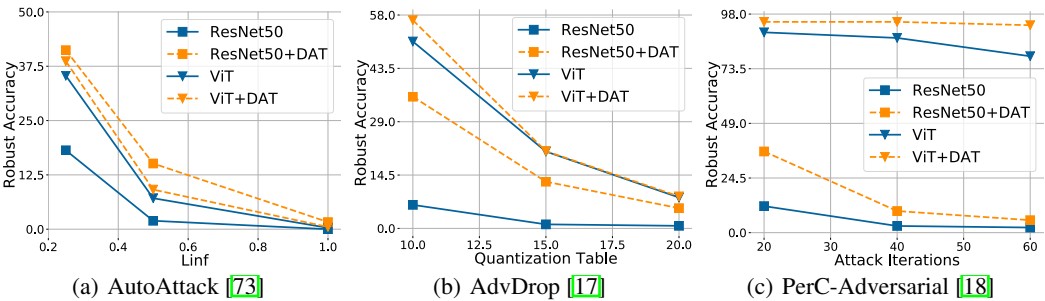

(a) AutoAttack [73]  (b) AdvDrop [17]  (c) PerC-Adversarial [18]

Figure 4: The adversarial robustness test under other stronger attackers.

## 4.5 In-depth Analysis

**Discrete Perturbations vs. Pixel-Space Perturbations** We compare the proposed Discrete Adversarial Examples (DAEs) with traditional Pixel-space Adversarial Examples (PAEs) in Figure 5. DAEs have following three superior properties: 1) DAEs are more realistic. By calculating the number of colors [17], we find PAEs add more invalid colors, resulting in a noisy image. While DAEs have minor changes on the color numbers, its modification is harder to be perceived by humans. To give quantitative results, we additionally report the FID score of DAEs. The FID score of our DAEs is 14.65, which is lower than 65.18 of AEs. The results are also consistent with our visual presentation result in Figure 5; 2) DAEs have less high frequency component compared with AEs in frequency analysis. It may lead the DAEs more close to natural image distributions; 3) Discrete perturbations are more structural. From the perturbation visualization in third row of Figure 5, we find discrete perturbations have more structured information about objects, shown it attends to more important locations. While pixel-wise perturbations are noisy and disordered.

**Visualized Attention.** We visualize the attention of ViT trained by DAT in Figure 6(b). For the object with unusual renditions in ImageNet-R, ViT cannot attend to semantically relevant image regions. While our DAT can locate the attention to the central object more related for the classification. This phenomenon is also reflected by the statistical average of the attention in Figure 6(c). By randomly sampling 1000 images in ImageNet-R and averaging the attention maps, we find the attentions of our DAT are more global. Compared with ViT which puts much attentions on the corners of the image found by prior work [20], our DAT additionally attends the central regions in where the classified object is often located.

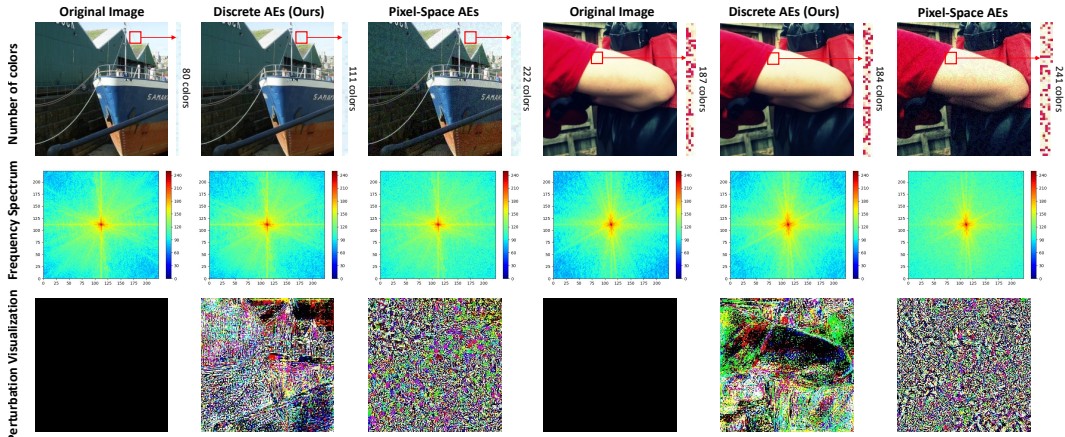

Figure 5: Comparison of discrete perturbations and pixel-space perturbations.

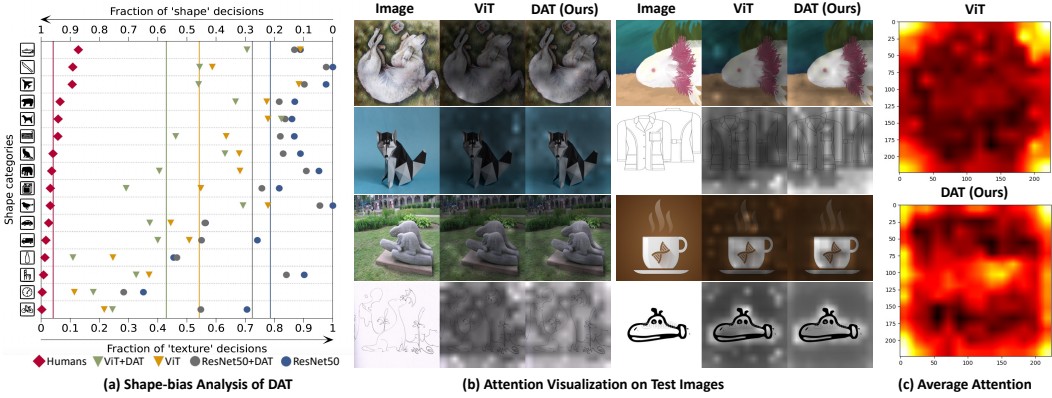

**(a) Shape-bias Analysis of DAT**

**(b) Attention Visualization on Test Images**

**(c) Average Attention**

Figure 6: (a) The fraction of correct shape-based decisions of models w/ and w/o DAT. (b) Visualized attention on test images of ImageNet-R. (c) The heat map of averaged attention with ViT and ViT trained by our DAT.

**Shape-bias Analysis.**    We conduct an analysis based on shape-bias, which represents the fraction of correct decisions based on object shape. The result is shown in Figure 6(a). The averaged scores on 16 categories is denoted by the colored vertical line. We compare decisions with Humans, ResNet50 w/ and w/o DAT, ViT w/ and w/o DAT. Human decisions are highly based on shape, which achieve the best average fraction of 0.96 to correctly recognize an image by shape. By comparison, ViT and ResNet50 still have large gap with humans on the ability of learning shape features. In this work, we find our DAT can help for improving the fraction of shape-based decisions of models. It suggests DAT regularizes the models to learn texture independent shape features, and behave more like a human.

# 5   Limitations and Conclusions

In this paper, we find transferring NLP-style adversarial training to vision models can enhance the learned visual representation effectively. We propose Discrete Adversarial Training (DAT), where images are presented as discrete visual words by VQGAN, and the model is training on examples which have the adversarially altered discrete visual representation. DAT needs not to modify the model architecture and works for both CNNs and ViTs across multiple tasks. DAT reports the state-of-the-art robustness on ImageNet-C and Stylized-ImageNet, exhibiting strong generalization. However, DAT still costs increased training time, this limitation also holds for any adversarial training. The strict assumption of an ideal discretizer is another potential limitation, despite DAT has used powerful VQGAN model to approach this assumption. The effect of DAT is empirically studied without deeper theoretical explanation. All the above limitations are remained as the future optimization direction.

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
