The Appendix is organized as follows. Appendix A extends the discussion on training of VQGAN. Appendix B presents more experimental results including comparison of robust training strategies, effect analysis on different perturbations, images corruptions and inference with the discretizer $\mathcal{Q}$. Besides, we analyse the training budget of our DAT. Appendix C discusses the implementation details of the three attackers in ablation experiments. Finally we do some visualizations in Appendix D.

## A  Details of Training VQGAN

We add more details about how VQGAN can be trained to discretize a continous image. The forward pass of VQGAN has been introduced in Sec 3.2.1. Suppose a reconstructed image $\hat{x}$, the training objective for VQGAN is defined between $\hat{x}$ and $x$ as:

$$\mathcal{L}_{\text{VQGAN}} = \min_{\text{Enc},\text{Dec},\mathcal{Z}} \max_{D} \mathbb{E}_{x \sim p(x)}[\mathcal{L}_{\text{VQ}}(\text{Enc}, \text{Dec}, \mathcal{Z}) + \mathcal{L}_{\text{GAN}}(\{\text{Enc}, \text{Dec}, \mathcal{Z}\}, D)] \tag{8}$$

$$\mathcal{L}_{\text{VQ}}(\text{Enc}, \text{Dec}, \mathcal{Z}) = \|x - \hat{x}\|_{\text{precept}} + \|\text{sg}[\text{Enc}(x)] - v_{\mathbf{q}}\|_2^2 + \|\text{sg}[v_{\mathbf{q}}] - \text{Enc}(x)\|_2^2 \tag{9}$$

$$\mathcal{L}_{\text{GAN}}(\{\text{Enc}, \text{Dec}, \mathcal{Z}\}, D) = [\log D(x) + \log(1 - D(\hat{x}))], \tag{10}$$

where $\|x - \hat{x}\|_{\text{precept}}$ is the perceptual reconstruction loss instead of $L_2$ loss, $\text{sg}[\cdot]$ denotes the stop-gradient operation, and $\|\text{sg}[v_{\mathbf{q}}] - \text{Enc}(x)\|_2^2$ is the commitment loss [38]. A patch-based discriminator $D$ is introduced for improving the quality of generated images by adversarial learning [74].

In this work, we use the pre-trained VQGAN weights for DAT directly, which are opened on GitHub[3,4]. VQGAN with $f = 8$, $d = 4$ and $K = 16384$ is used for main experiments. We acknowledge the authors of [13, 42], whose works have greatly promoted our DAT.

## B  More Experimental Results

### B.1  Strategies Comparison on ResNet50

Table 6 presents the comparison results of DAT with other robust training strategies on ResNet50. Note that Advprop, Fast Advprop and Debiased use auxiliary BN while Pyramid AT and DAT not. Advprop achieves the best clean accuracy. The reason may lie in the auxiliary BN reduces the impact of adversarial examples on standard performance. On ResNet50, DAT is not always the best on the robustness benchmarks. By augmenting with the style transferred data, Debiased achieves best on IN-C, IN-Sketch and Stylized IN. On the contrary, such robust training method proposed for ViTs, e.g. Pyramid AT, is not incompatible with ResNet50 and obtain unsatisfactory results. However, our DAT is a general method which works for both CNNs and ViTs, and can be more practical.

| Training Strategies | ImageNet | Adversarial Robustness | | Out of Distribution Robustness | | | | | |
|---|---|---|---|---|---|---|---|---|---|
| | | FGSM | DamageNet | A | C↓ | V2 | R | Sketch | Stylized |
| Normal [48] | 76.13 | 12.19 | 5.94 | 0 | 76.70 | 63.2 | 36.17 | 24.09 | 7.38 |
| Advprop [26] | **77.59** | 28.65 | **15.58** | 4.33 | 70.53 | **65.47** | 38.75 | 25.51 | 7.99 |
| Fast Advprop [27] | 76.6 | 17.33 | 7.45 | 2.19 | 73.31 | 64.24 | 38.17 | 25.03 | 8.3 |
| Pyramid AT [28] | 75.46 | 30.35 | 14.22 | 3.01 | 76.42 | 62.46 | 38.85 | 23.76 | 10.41 |
| Debiased [49] | 76.91 | 20.4 | 6.66 | 3.51 | **67.55** | 65.04 | 40.8 | **28.42** | **17.4** |
| **DAT (Ours)** | 76.52 | **30.66** | 14.42 | **4.38** | 74.16 | 65.02 | **41.9** | 27.27 | 10.8 |

Table 6: Comparison of DAT with other training strategies. We use ResNet50 as the base model.

### B.2  Comparing with Traditional Adversarial Training

We compare our DAT with open-sourced adversarially robust models[5] in Table 7. From the results, we can see a clear quantification of the benefit of our proposed discrete AT scheme compared with traditional AT. For clean performance, traditional AT plays a negative impact. However DAT can reduce the negative impact and achieve higher accuracy on validation set of ImageNet. It even surpasses the clean performance of normal training. The results also suggest AT with a very small $\epsilon$

---

[3]https://github.com/CompVis/taming-transformers
[4]https://github.com/CompVis/latent-diffusion
[5]https://github.com/microsoft/robust-models-transfer

can slightly benefit the generalization, e.g., with $\epsilon$=0.01 L2 AT, ImageNet-C mCE value from 76.70 drops to 75.33. But with the $\epsilon$ becoming larger, AT greatly damages the generalization, e.g. with $\epsilon$=5.0 L2 AT, ImageNet-C mCE value increases to 88.98. In contrast, our DAT achieves significant improvement on generalization compared with traditional AT.

| Models | Train Cost | ImageNet | A | C↓ | V2 | R | Sketch | Stylized |
|---|---|---|---|---|---|---|---|---|
| Normal training, $\epsilon$=0 | 1× | 76.13 | 0.0 | 76.70 | 63.20 | 36.17 | 24.09 | 7.38 |
| L2-Robust, $\epsilon$=0.01 | 4× | 75.68 | 2.11 | 75.33 | 64.00 | 35.98 | 23.55 | 7.47 |
| L2-Robust, $\epsilon$=0.03 | 4× | 75.76 | 2.17 | 75.36 | 63.66 | 36.18 | 23.98 | 8.18 |
| L2-Robust, $\epsilon$=0.05 | 4× | 75.59 | 2.19 | 75.65 | 63.37 | 36.48 | 23.90 | 8.51 |
| L2-Robust, $\epsilon$=0.1 | 4× | 74.78 | 2.13 | 75.42 | 62.64 | 36.90 | 23.85 | 9.18 |
| L2-Robust, $\epsilon$=0.25 | 4× | 74.14 | 2.28 | 75.79 | 62.20 | 37.57 | 24.33 | 10.07 |
| L2-Robust, $\epsilon$=0.5 | 4× | 73.16 | 2.19 | 75.91 | 60.48 | 38.03 | 23.49 | 10.99 |
| L2-Robust, $\epsilon$=1.0 | 4× | 70.43 | 2.19 | 78.36 | 57.36 | 38.21 | 22.63 | 11.07 |
| L2-Robust, $\epsilon$=3.0 | 4× | 62.83 | 1.97 | 83.84 | 49.45 | 36.48 | 20.40 | 10.48 |
| L2-Robust, $\epsilon$=5.0 | 4× | 56.13 | 1.71 | 88.98 | 43.04 | 32.75 | 16.82 | 9.13 |
| Linf-Robust, $\epsilon$=0.5/255 | 4× | 73.73 | 2.35 | 76.86 | 61.88 | 38.54 | 23.79 | 10.94 |
| Linf-Robust, $\epsilon$=1.0/255 | 4× | 72.05 | 2.53 | 78.34 | 59.60 | 40.13 | 23.70 | 12.10 |
| Linf-Robust, $\epsilon$=2.0/255 | 4× | 69.10 | 2.52 | 80.09 | 56.64 | 38.65 | 22.14 | 12.36 |
| Linf-Robust, $\epsilon$=4.0/255 | 4× | 63.86 | 2.25 | 85.14 | 51.39 | 38.25 | 20.94 | 11.70 |
| Linf-Robust, $\epsilon$=8.0/255 | 4× | 54.53 | 2.12 | 91.59 | 42.16 | 34.40 | 18.10 | 9.58 |
| DAT (Ours) | 3.5× | 76.52 | 4.38 | 74.16 | 65.02 | 41.90 | 27.27 | 10.8 |

Table 7: Comparison of our DAT with adversarial training models.

## B.3 DAT with Different Perturbations

Table 8 counts for the percentage of the modified visual words with different magnitude $\alpha$. We show only 3.8% of the visual words are changed when we set $\alpha$ as 0.1. This proportion does not have negative impact on the trained models, but even somewhat benefits the clean accuracy. With larger proportion of visual words being adversarially altered, the standerd performance and generalization are getting worse. We also experiment DAT with random perturbations. To introducing the randomness, we selects 3.8% visual words and replaces them with other words. We find this operation can slightly improvement the generalization of learned representation, but it still cannot achieve the comparable effect as our DAT. It suggests adversarially altering the visual words is a better way.

| Types | $\alpha$ | Modified Codes | ImageNet | Adversarial Robustness | | Out of Distribution Robustness | | | | | |
|---|---|---|---|---|---|---|---|---|---|---|---|
| | | | | FGSM | DamageNet | A | C↓ | V2 | R | Sketch | Stylized |
| Random | - | 3.8% | 76.47 | 29.01 | 10.7 | 3.00 | 74.71 | 64.75 | 40.19 | 26.17 | 9.89 |
| Adv. | 0.0 | 0.0% | 76.38 | 23.94 | 9.12 | 3.2 | 76.31 | 64.71 | 38.41 | 24.62 | 8.77 |
| Adv. | 0.1 | 3.8% | 76.52 | 30.66 | 14.42 | 4.38 | 74.16 | 65.02 | 41.9 | 27.27 | 10.8 |
| Adv. | 0.2 | 7% | 75.93 | 34.47 | 15.21 | 3.11 | 75.09 | 64.38 | 40.27 | 26.33 | 10.14 |
| Adv. | 0.4 | 13% | 74.28 | 36.2 | 17.76 | 1.96 | 77.25 | 62.5 | 38.75 | 24.31 | 8.61 |

Table 8: DAT with different perturbations. We use ResNet50 as the base model.

## B.4 The Effect of DAT on Corruptions in ImageNet-C

To analyse the effect of DAT on each image corruption in ImageNet-C, we report the detailed results in Table 9. For ResNet50, we find DAT reduces the accuracy on images with contrast and fog corruptions. It demonstrates that ResNet50 trained by DAT can be sensitive to the image lack of the hierarchy. However, for ViT, DAT can improve the performance on all corruptions. It suggests DAT works more efficiently on transformer-based vision models.

| Model | Average | Blur | | | | Noise | | | Digital | | | | Weather | | | |
|---|---|---|---|---|---|---|---|---|---|---|---|---|---|---|---|---|
| | | Motion | Defoc | Glass | Zoom | Gauss | Impul | Shot | Contr | Elast | JPEG | Pixel | Bright. | Snow | Fog | Frost |
| ResNet50 | 39.2 | 38.7 | 38.8 | 26.8 | 36.2 | 29.2 | 23.8 | 27.0 | 39.1 | 45.3 | 53.4 | 44.8 | 68.0 | 32.5 | 45.8 | 38.1 |
| +DAT (Ours) | 41.1 | 38.3 | 37.2 | 33.7 | 37.9 | 33.0 | 28.1 | 31.1 | 36.5 | 50.0 | 59.0 | 45.6 | 69.0 | 34.2 | 41.5 | 41.0 |
| ViT | 57.2 | 54.2 | 47.9 | 43.0 | 41.6 | 61.9 | 58.4 | 58.3 | 60.2 | 58.0 | 61.4 | 65.9 | 74.8 | 52.1 | 61.6 | 59.5 |
| +DAT (Ours) | 65.2 | 58.4 | 55.1 | 49.8 | 50.8 | 71.3 | 70.3 | 70.5 | 71.7 | 63.1 | 69.1 | 67.7 | 78.2 | 64.0 | 69.6 | 68.2 |

Table 9: Detailed results of DAT on each image corruption in ImageNet-C.

## B.5 Training Budget for DAT

DAT is experimented on 32 2080Ti GPUs. We compare the training cost with other robust training strategies in same setting. The results is shown in Table 10. DAT only needs one attack step to generate discrete adversarial examples, which makes it less expensive than standard adversarial training. However, DAT still requires $3.5\times$ training budget than normal training. We believe that reducing the cost of DAT is necessary, which will be remained as the future work.

| Training Strategies | Attack Steps | Training Budget |
|---|---|---|
| Normal | 0 | $1\times$ |
| Adversarial Training | 10 | $11\times$ |
| Advprop | 5 | $7\times$ |
| Advprop | 1 | $3\times$ |
| DAT (Ours) | 1 | $3.5\times$ |

Table 10: Comparison of the training costs.

## B.6 Inference with the Discretizer $\mathcal{Q}$

In this work, we delete the discretizer $\mathcal{Q}$ at inference time. However, there is another option that remaining the discretizer for test inputs. To study the effect of this alternative, we report some results in Table 11. Although inference with $\mathcal{Q}$ brings improvement on adversarial robustness, it meanwhile harms the standard performance and generalization. The inference cost also increases by the additional computation on $\mathcal{Q}$. Therefore, such alternative cannot yield the best trade-off on speed and performance, which is not adopted by our DAT.

| Methods | ImageNet | Adversarial Robustness | | Out of Distribution Robustness | | | | | |
|---|---|---|---|---|---|---|---|---|---|
| | | FGSM | DamageNet | A | C$\downarrow$ | V2 | R | Sketch | Stylized |
| ResNet50 + DAT (w/o $\mathcal{Q}$) | 76.52 | 30.66 | 14.42 | 4.38 | 74.16 | 65.02 | 41.90 | 27.27 | 10.8 |
| ResNet50 + DAT (w/ $\mathcal{Q}$) | 74.8 | 55.4 | 20.44 | 4.13 | 76.06 | 63.04 | 39.67 | 25.64 | 10.16 |
| ViT + DAT (w/o $\mathcal{Q}$) | 81.46 | 51.82 | 45.70 | 30.15 | 44.65 | 70.83 | 47.34 | 34.77 | 23.13 |
| ViT + DAT (w/ $\mathcal{Q}$) | 80.12 | 61.65 | 50.4 | 22.44 | 48.43 | 68.59 | 47.2 | 34.41 | 21.83 |

Table 11: The ablation on the discretizer $\mathcal{Q}$ at inference time.

## B.7 Is it necessary for bounding $\delta$?

Traditional adversarial attacks always add constraints on perturbations. While in this work, there are no restrictions on $\delta$. To explore if bounding $\delta$ is necessary in DAT, we add $l_\infty$ bound on $\delta$ with different $\epsilon$ and re-run the DAT. The result is shown in Table 12. DAT achieves best performance when $\delta$ is not bounded. The worst result is appeared when $\delta$ is bounded with $\epsilon = 1/255$. With larger $\epsilon$, the results become better. Therefore, it seems bounding $\delta$ in our DAT is not necessary, and even plays negative affect on the overall performance.

| $\epsilon$ of $l_\infty$ | ImageNet | Adversarial Robustness | | Out of Distribution Robustness | | | | | |
|---|---|---|---|---|---|---|---|---|---|
| | | FGSM | DamageNet | A | C$\downarrow$ | V2 | R | Sketch | Stylized |
| 1/255 | 76.10 | 29.41 | 12.00 | 3.53 | 75.53 | 64.11 | 39.05 | 25.04 | 8.69 |
| 2/255 | 76.16 | 29.75 | 13.24 | 3.75 | 74.87 | 64.32 | 40.38 | 25.53 | 9.31 |
| 4/255 | 76.47 | 31.43 | 14.25 | 4.31 | 74.12 | 65.07 | 41.68 | 26.99 | 10.62 |
| $\infty$ (Ours) | 76.52 | 30.66 | 14.42 | 4.38 | 74.16 | 65.02 | 41.90 | 27.27 | 10.8 |

Table 12: The trained model on DAT when $\delta$ is bounded by different $\epsilon$.

## B.8 DAT for Domain Generalization

In addition to training on large-scale ImageNet, we evaluate our DAT on Domain Generalization (DG) tasks. DG task is more challenging since it needs the learned model to transfer between multiple domains, using only small amount of the data. We adopt PACS dataset, which consists of four domains, namely Photo, Art Painting, Cartoon and Sketch. Each domain contains seven categories. For each trial, we train on 3 domains for generalizing to remaining unseen domain. To keep the setting consistent with previous works, we use AlexNet as the backbone. The results are shown in

below Table 13. We only compare with previously adversarial augmentation based DG methods. DAT can also achieve better domain generalization performance on PACS dataset. It has slight drop on domain of photo, but improves the transferability on other three domains.

| | ADA | MD-ADA | DAT (Ours) |
|---|---|---|---|
| Art Painting | 64.3 | 67.1 | **67.3** |
| Cartoon | 69.8 | 69.9 | **71.3** |
| Photo | 85.1 | **88.6** | 87.8 |
| Sketch | 60.4 | 63.0 | **64.1** |
| Average | 69.9 | 72.2 | **72.6** |

Table 13: Classification accuracy (%) of our DAT on PACS dataset in comparison with the previously adversarial augmentation based DG methods.

## C  Implementation of the Stronger Attacks in Ablation Experiments

For AutoAttack, we attack a subset of ImageNet provided in RobustBench [6], which consists of 5000 images. Three perturbations bounded with $l_\infty$-norm of $1/255$, $0.5/255$ and $0.25/255$ are adopted. For AdvDrop, the test dataset is 1000 random sampled images on ImageNet, which is provided in the official implementaton [7]. The bound of the quantization table is a key factor in AdvDrop which controls the attack strength. We use three bounds with 10, 15 and 20 to regularize the quantization table. For PerC-Adversarial, we use the Perceptual Color distance Alternating Loss (PerC-AL) method to generate adversarial examples. PerC-Adversarial uses the test data of Defense Against Adversarial Attack Challenge in NeurIPS 2017. We change the attack strength by three different attack iterations: 20, 40 and 60.

## D  Visualization

### D.1  Comparison of the Reconstruction Quality

We compare the reconstruction quality of different discretizers $\mathcal{Q}$ in Figure 7. VQGAN with $k = 16384$ and $f = 8$ can retain the most of the image details, which has the least impact on the classification models when used for training. With the growing of the downsampling factor $f$, some fine-grained attributes of the objects can be changed. For example, in third row the spotted texture on the peacock's tail is partly lost after reconstruction. Compared with VQGAN models, DALL-E blurs the image to a greater extent, yielding a low quality reconstructed image. Accordingly, DAT which uses DALL-E for image discretization performs worst on clean accuracy in Table 5. It shows that the reconstruction quality is indeed proportional to the standard performance in DAT.

### D.2  Visualization of Discrete Adversarial Examples

We visualize the discrete adversarial examples which is generated by DAT for training in Figure 8. With the growing of $\alpha$, the alteration on images become larger. However, different from traditional adversarial example which essentially adds global high-frequency noise on images, we find DAT modifies the properties of the local part of the object. For example, in the fifth row of Figure 8, the discrete adversarial example is changing the eye color of the cat. Such modification is large but imperceptible and semantic-preserving. There are also some failure case where DAT changes the semantics. In sixth row of Figure 8, after discrete reconstruction, the otter in image looks more like a dog. We believe that such cases are only a minority, and will not affect the overall performance of DAT.

### D.3  Visualization of the Straight-Through Gradients and Backward Gradients in DAT

We visualize the estimated straight-through gradients in Eq 7 and the directly backward gradients in Eq 6. As shown in Figure 9, the approximated gradient by straight-through estimator does accurately

---

[6]https://github.com/RobustBench/robustbench
[7]https://github.com/RjDuan/AdvDrop

Original VQGAN_k16384_f8 VQGAN_k256_f8 VQGAN_k16384_f16 VQGAN_IN_k16384_f16 DALL-E

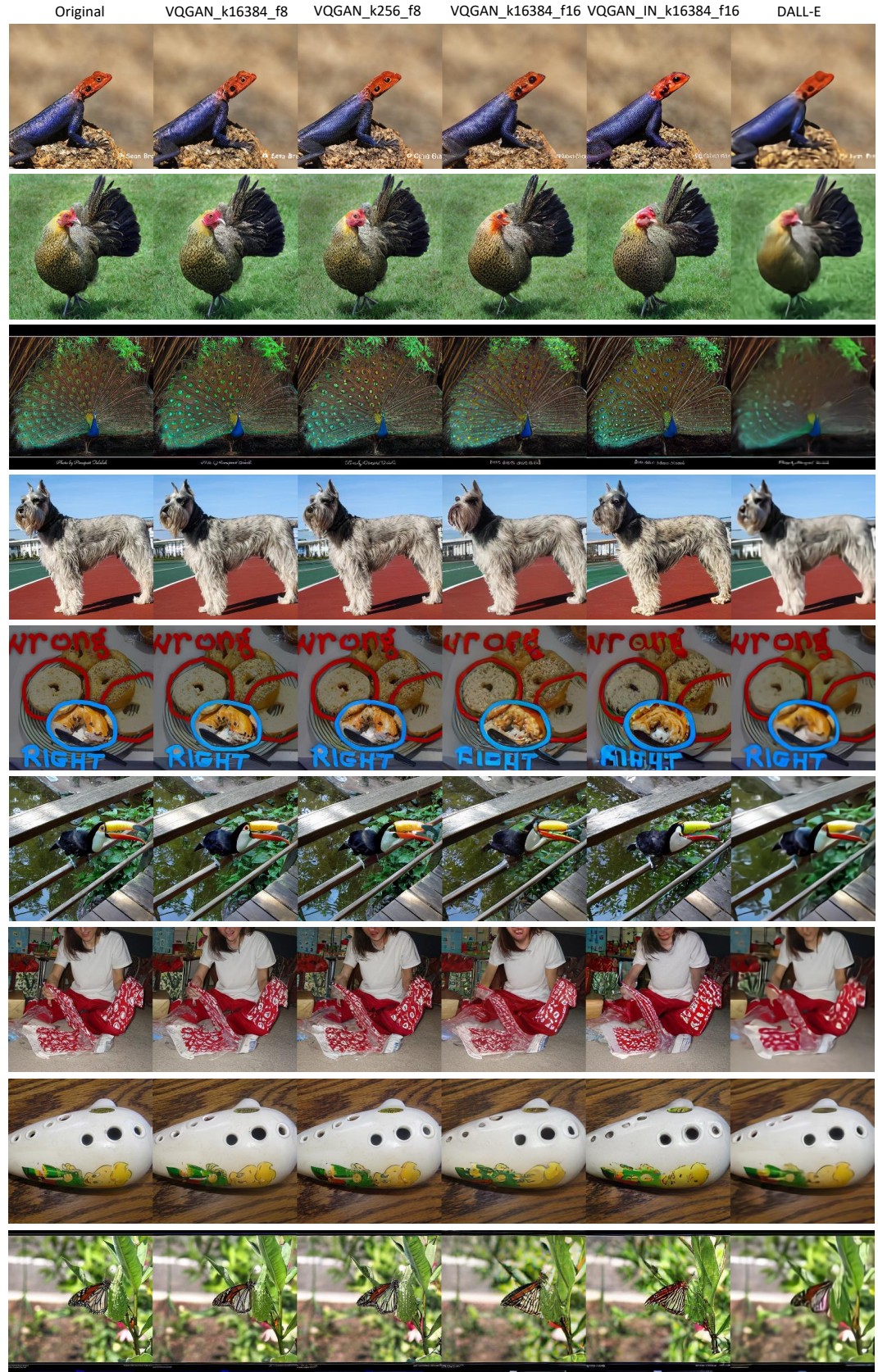

Figure 7: The visualization of the discrete reconstruction $\hat{x}$ based on different discretizers.

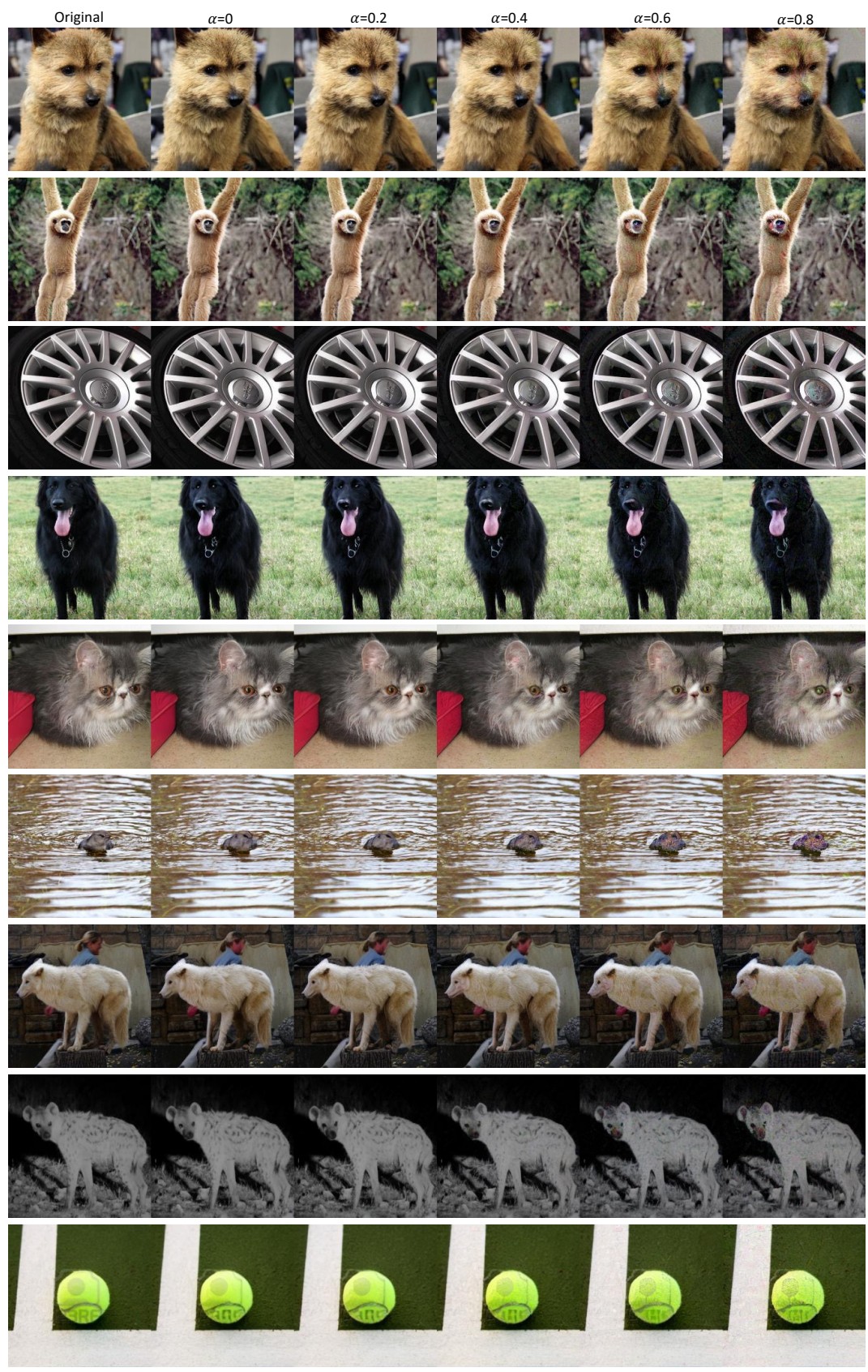

Figure 8: Training example visualization of DAT with different $\alpha$.

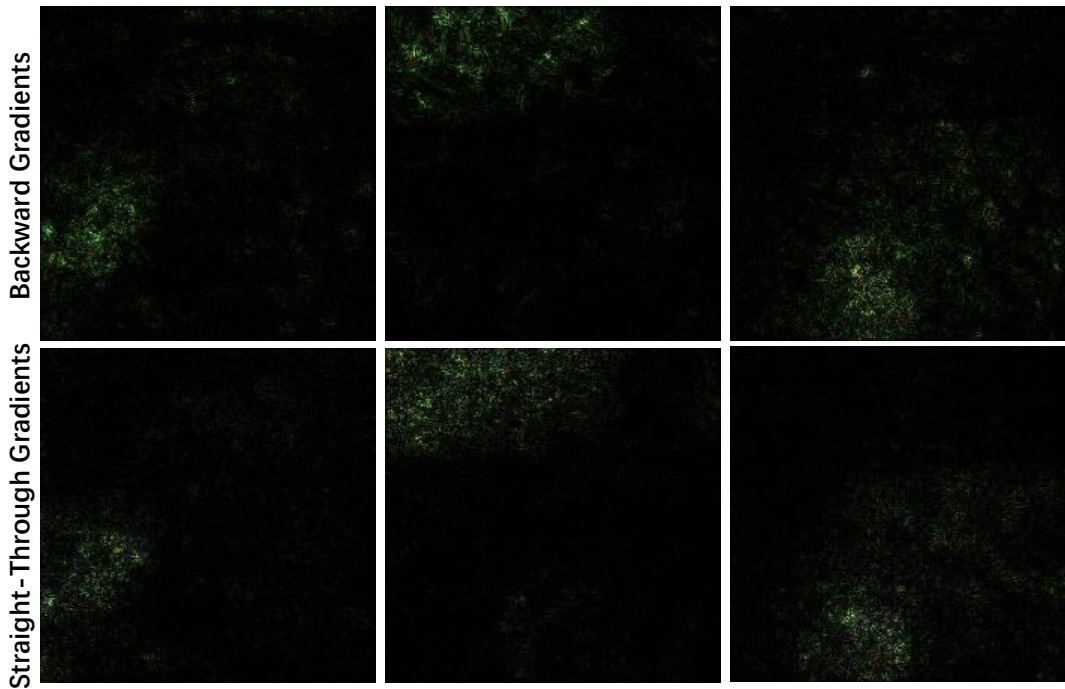

Figure 9: Visualization of the straight-through gradients and backward gradients.

estimate the ground-truth gradients in Eq 6. It reflects the rationality of the proposed efficient straight-through gradient in DAT.