# OpenReview forum: "Enhance the Visual Representation via Discrete Adversarial Training"
_NeurIPS.cc/2022/Conference — NeurIPS 2022 Accept_

### Official Review · Reviewer_C7HU · 2022-07-06

**Rating:** 6
**Confidence:** 5
**Soundness:** 3 good
**Presentation:** 3 good
**Contribution:** 3 good

**Summary:**

This paper studies the problem of adversarial training for vision tasks by borrowing ideas from NLP.  A new method called "DAT" (Discrete Adversarial Training) is proposed.  DAT uses a VQA-GAN encoder to convert images into discrete tokens, by learning a vocabulary of words (aka image codebook).  Adversarial perturbations are applied on these tokens.  Performance is reported on:
1. 6 versions of Imagenet for evaluating robustness (Imagenet-A/C/V2/R/ Stylized/Sketch)
2. against adversarial attacks (FGSM, DamageNet)
3. COCO-C for object detection

**Questions:**

## Questions
Adversarial training has also been used in image classification literature for domain generalization:  see (GUD: Volpi et al NeurIPS 2018 https://arxiv.org/abs/1805.12018, MADA: Qiao et al. CVPR 2019 https://arxiv.org/abs/2003.13216, ME-ADA: Zhao et al. NeurIPS 2020 https://arxiv.org/abs/2010.08001). This leads me to two questions:
1. How does DAT perform on domain generalization benchmarks such as Digits, PACS, OfficeHome etc.?
2. On the imagenet benchmarks presented in this paper, how does DAT compare with GUD, MADA? (if some of the baselines that you have used have been shown to be better than GUD, MADA -- please let me know which paper).

3. Line 148: `We delete the constraint term since there is no need to bound the per-pixel values of $\delta$` -- what are the effects on the performance if the perturbation is bounded? is it better / worse in general? Is is better for some eval datasets but worse for others?
4. Line 166: `Since xˆ ≃ x established for an ideal discretizer Q` -- this connects to my question-3:  does the magnitude of the perturbation affect this assumption? Will a large perturbation make this assumption false?
5. Line 184: `Previous work has pointed out that the underlying distributions of adversarial examples are different from clean images.` -- which paper? please cite.

## Suggestions (about References, Structure, Grammar, etc.)
1. References:  you may also add references to "vision-like" AT that have been explored in NLP.  You have mentioned one of them (FreeLB). Another one is:
    - https://arxiv.org/abs/2006.06195 (NeurIPS 2020) -- here "VILLA" perturbs encoded features (of images and/or text) encoded by pretrained V&L model.  In that sense, it is similar to perturbing the VQGAN encoding of images (DAT) is similar to VILLA, but DAT perturbs in the symbolic codebook space.
2. References:  other forms of adversarial training (beyond pixel-wise perturbations) have been explored for vision tasks:
    - https://arxiv.org/abs/2012.01806 (AAAI 2021) -- "AGAT" is an adversarial training pipeline which starts with a given knowledge of symbolic attributes of images, and then perturbs images along those attributes.  A comparison should be made between DAT and AGAT: in my view DAT indirectly uses the symbolic knowledge from VQGAN encoder, whereas AGAT assumes that such symbols/attributes will be given.
3. Typo: Fig 4 caption:  last word should be "attackers".
4. Fig 3 caption: please mention in words what $\alpha$ is (step size/magnitude of adversarial update).
5. Fig 2: please label the x-axis
6. Line 201: `Using robustness`:  please rephrase this to `using "robustness" library` since robustness is a standard ML term, so people might get confused.

**Limitations:**

Limitations are not discussed clearly.  Please add a section in/after conclusion to discuss them.  Especially about the assumptions in the method.

**Strengths And Weaknesses:**

## Strengths
1. The motivation of the paper comes from an informed perspective of both vision and NLP literature on AT, and I like the fact that the phenomenon from NLP of AT improving both robustness and clean accuracy is used as a goal for AT in vision.  This exchange of ideas often leads to improvements across various application domains of machine learning.
2. Performance is evaluated extensively and reported
    - 3 benchmarks (classification, detection, self-supervised image classification)
    - multiple architectures
3. The paper is well written, especially the methods section.  I have a few questions about the method though (see below)

## Weaknesses
1. One major weakness is that previous work on adversarial training has been used for improving domain generalization (see my question1).  However, experiments have not been performed in this paper on DG benchmarks.
2. The methods section has some assumptions (see Q 3 and 4) -- these assumptions are not clearly justified (either theoretically or intuitively).  More study along these assumptions is needed.
3. The effect of discrete perturbations is not discussed.  In pixel-wise AT, the perturbation leads to noisy images.  But that need not be true for DAT -- a comparison should be included, perhaps in terms of photorealism comparison between "real" images and perturbed images.

---

> ### Author Response · Authors · 2022-08-02
> **Response to Reviewer C7HU (1 of 2)**
>
>
> We thank the reviewer for the time and insightful comments.
>
> **Weakness3:** The effect of discrete perturbations is not discussed. In pixel-wise AT, the perturbation leads to noisy images. But that need not be true for DAT -- a comparison should be included, perhaps in terms of photorealism comparison between "real" images and perturbed images.
>
> **Reply:** An advanced property of discrete perturbations is that it does not change the distribution of the original image in most case. As discussed in Line 182, such property reduces the distribution shift between clean images and AEs in traditional AT, yielding both robustness and clean performance improvement.
>
> To explore what effect of discrete perturbations contributed to this advanced property, and what is the difference of discrete perturbations with pixel-wise perturbations, we add more discussion in Appendix D.5. For comparative fairness, we keep the same attack success rate of the generated discrete and pixel-wise AEs. The study has three aspects:
>
> - *Discrete perturbations create more realistic AEs.* We add a visualization of pixel-wise AEs and discrete AEs in Figure 8 of Appendix D.5 for subjective photorealism comparison. As said for the reviewer, pixel-wise perturbations lead to noisy images. By calculating the number of colors [17], we find pixel-wise AEs add more  invalid colors, resulting in a noisy image. While discrete perturbations have minor changes on the color numbers of original image. Such subtle change is hard to be perceived by humans.
> - *Discrete perturbations have more low frequency component.* We conduct frequency analysis on compared AEs in second row of Figure 8. Pixel-wise perturbations introduce more high frequency component. It may lead the pixel-wise AEs to far away from natural distributions. However discrete perturbations will not introduce unnecessary high-frequency components in original image.
> - *Discrete perturbations are more structural.* From the perturbation visualization in third row of Figure 8 , we find discrete perturbations have more structured information about objects, shown it attends to more important locations. While pixel-wise perturbations are noisy and disordered.
>
> ------
>
> **Q1, Q2:** How DAT performs on DG tasks? if some of the baselines that you have used have been shown to be better than GUD, MADA?
>
> **Reply:** Thanks for your good advice. Results on Domain Generalization (DG) can demonstrate the ability of our DAT on small-scale tasks rather than ImageNet. It helps to further improve the significance of DAT.
>
> We carefully read the recommended papers: GUD, MADA, ME-ADA. Fortunately, we find on CIFAR-10-C benchmark, a part of compared baselines in this work have been shown to be better than GUD, MADA and ME-ADA. Specifically, the GUD, MADA and ME-ADA get 58.26%, 65.59%, and 80.5% average accuracy on CIFAR-10-C respectively. While in our paper, the compared baseline Augmix [32] achieves 87.5% avg. accuracy, surpassing above three methods with a large gap.
>
> But we think it is still necessary to evaluate DAT on DG benchmarks. We choose PACS datasets, and train on 3 domains for generalizing to remaining unseen domain. Standard AlexNet is used to keep the setting consistent with previous works. The results are shown in below table:
>
> | DomainID  | ME-ADA | DAT |
> |  ----  | ----  |  ----  |
> | Art | 67.1% | 67.3 |
> | Cartoon | 69.9% | 71.3 |
> | Photo | 88.6% | 87.8 |
> | Sketch | 63.0% | 64.1% |
> | Avg | 72.2% | 72.6% |
>
> DAT can also achieve better performance on DG task such as PACS. It has slight drop on domain of photo, but improves the transferability on other three domains.

---

> > ### Author Response · Authors · 2022-08-02
> > **Response to Reviewer C7HU (2 of 2)**
> >
> > **Q3:**  "We delete the constraint term since there is no need to bound the per-pixel values of $\delta$" --what are the effects on the performance if the perturbation is bounded? is it better / worse in general? Is is better for some eval datasets but worse for others?
> >
> > **Reply:** A similar question is proposed by Reviewer wZbH, where we explain why we do not bound the per-pixel values of $\delta$. Here we address your concerns about the effects on the performance if the perturbation is bounded. We choose $l_{\infty}$ bound to conduct some analytical experiments as follows:
> >
> > **(1) How per-pixel bound effects on $\delta$.** To study the effect of per-pixel bound on $\delta$, we first use $\alpha=0.1$ to generate 1000 samples and count the proportion of the $\delta$  in different perturbation intervals:
> > | Intervals  | Proportion of the $\delta$ in the interval |
> > |  ----  | ----  |
> > | (-1/255, 1/255) | 28.3% |
> > | (-2/255, 2/255) | 95.9% |
> > | (-4/255, 4/255) | 100% |
> >
> > With above table, using bound of (-4/255, 4/255) almostly has no impact of $\delta$. While using bound of (-1/255, 1/255) and (-2/255, 2/255) may potentially impact the training performance, which is discussed in below experiment.
> >
> > **(2)  How per-pixel bound effects on the performance of DAT.** We rerun DAT on ResNet50 by adding different $l_{\infty}$ bound on $\delta$, and keeping other settings unchanged.
> > | $l_{\infty}$ Bounds on $\delta$ | ImageNet-Val | FGSM | DamageNet | A | C↓| V2 | R | Sketch | Stylized |
> > |  ----  | ---- | ---- | ---- | ---- | ---- | ---- | ---- | ---- | ---- |
> > | no bounds  | 76.52 | 30.66 | 14.42 | 4.38 | 74.16 | 65.02 | 41.90 | 27.27 | 10.8 |
> > | (-4/255, 4/255) | 76.47 | 31.43 | 14.25 | 4.31 | 74.12 | 65.07 | 41.68 | 26.99 | 10.62 |
> > | (-2/255, 2/255) | 76.16 | 29.75 | 13.24 | 3.75 | 74.87 | 64.32 | 40.38 | 25.53 |9.31 |
> > | (-1/255, 1/255)| 76.10 | 29.41 | 12.00 | 3.53 | 75.53 | 64.11 | 39.05 | 25.04 | 8.69 |
> >
> > As shown in above table, DAT achieves best performance when $\delta$ is not bounded. The worst result is appeared when $\delta$ is bounded between (-1/255, 1/255). With larger $l_{\infty}$ bound, the results become better.
> >
> > ------
> >
> > **Q4:** Line 166: Since xˆ ≃ x established for an ideal discretizer Q -- this connects to my question-3: does the magnitude of the perturbation affect this assumption? Will a large perturbation make this assumption false?
> >
> > **Reply:** To assist our response of Q3, here we give some intuitive results to show the proximity between $Q(x)$ and $x$. We choose 5 different magnitudes by setting different $\alpha$ (0.0, 0.1, 0.2, 0.3, 0.4), and compute the L2 distance, LPIPS between original image and its corresponding reconstruction.
> >
> > |  Alpha  |  Interval of per-pixel value of $\delta$  |  L2  |  LPIPS  |
> > |  ----  | ----  | ----  | ----  |
> > |  0.0 |  -  | 5.4e-3 | 0.082 |
> > |  0.1 |  (-1.14/255, 1.14/255) |  5.4e-3  | 0.083 |
> > |  0.2 |  (-2.29/255, 2.28/255) |  5.4e-3  | 0.083 |
> > |  0.3 |  (-3.44/255, 3.41/255) |  5.4e-3  | 0.083 |
> > |  0.4 | (-4.58/255, 4.57/255) |  5.4e-3  | 0.083 |
> >
> > When $\alpha=0.0$, $Q(x)$ has a very small L2 distance to original $x$. It suggests the strong reconstruction ability of VQGAN model, which greatly supports our assumption. Meanwhile, adding larger perturbation on $x$ has little effect on the reducibility of the reconstruction process. Our assumption is not false under such condition.
> >
> > ------
> >
> > **Q5:** Line 184: Previous work has pointed out that the underlying distributions of adversarial examples are different from clean images. -- which paper? please cite.
> >
> > **Reply:** Sorry for the unclarity, we have added the citation ([26]) in revised paper.
> >
> > ------
> >
> > **Suggestions:** add discuss about VILLA and AGAT
> >
> > **Reply:**  Thanks for your kind suggestion, we have discussed the VILLA and AGAT in the Related Work of the revised version. VILLA is also a representation enhancement technique using AT, but it is only applied for vision-and-language representation learning task. AGAT relies on a set of pre-defined attributes, this constraint makes it hard to transfer broader tasks where attributes are not given. As compared, DAT is more generic to most vision tasks, and it does not require any additional attribute annotation.
> >
> > ------
> >
> > **Suggestions:** other typos
> >
> > **Reply:** Thanks very much for pointing out the typos in our paper. We really appreciate you for your carefulness and conscientiousness. We have carefully checked the typos and improved the writing in the revised manuscript.
> >
> > ------
> >
> > **Limitations:** add discussion of limitations
> >
> > **Reply:** We have discussed the limitation in section of Conclusions. See in revised paper.
> >
> > ------
> >
> > We hope that our response has addressed all of your concerns. Thank you for your time and feedback on our submission! Please don't hesitate to let us know if you have any remaining questions or concerns.

---

> > > ### Comment · Reviewer_C7HU · 2022-08-07
> > > **Response to rebuttal**
> > >
> > > I appreciate the detailed response.  Here are my thoughts:
> > >
> > > - **Weakness 3:** Fig 8 is useful, thanks for adding it.  The points raised about low-frequency vs high-frequency noise and "invalid colors" is also useful. However I don't see a photorealism comparison still.  One way to achieve this would be to compute FID score between distribution of real images and distribution of DAT attacked images ; and compare it with FID of real vs pixel-level.  See this library for a quick implementation of FID https://github.com/mseitzer/pytorch-fid
> > >
> > > - **Q1/Q2/Weakness 1 (about DG):** Thank you for the PACS experiment.  The table for ME-ADA vs DAT is useful.  In the final version of the paper, I would encourage authors to also add these results (and if time permits, also replicate it on other datasets, for eg. the benchmark of Table 1 in ME-ADA paper).  This will strengthen the evidence for the efficacy of DAT, and make a larger impact on readers interested in robustness, DG, AT, etc.
> > >
> > > - **Q3  and Q4** - Thanks for the answers and supporting experiments.  For Q4 you may rephrase the statement to something like "it is empirically observed that $\hat{x} \sim x $ ... " rather than saying "it is established".
> > >
> > > - **Q5/references/typos/limitations**  thanks for the update.
> > >
> > > Overall comment:
> > > *My original rating was Weak Accept (6).  Based on your response, I am inclined to increase my rating.*

---

> > > > ### Author Response · Authors · 2022-08-07
> > > > **Further response of the remaining questions**
> > > >
> > > > Thank you for your feedback. We are very happy to see that most of your concerns have been resolved, and many thanks to the reviewer for helping us improve the paper.
> > > >
> > > > **(1) Reply of Weakness 3:**  We appreciate the good advice of FID as the photorealism comparison. The results of FID score w.r.t. clean examples vs. discrete AEs and clean examples vs. pixel-level AEs are shown in below table:
> > > > | Settings | FID score |
> > > > |  ----  | ----  |
> > > > | original input vs. reconstructed images | 1.14 |
> > > > | original input vs. pixel-level AEs | 65.18 |
> > > > | original input vs. discrete AEs | 14.65 |
> > > >
> > > > For fairness, we guarantee that all compared AEs have same attack ability. The result suggests our discrete AEs are more photorealistic than traditional pixel-level AEs. It is also consistent with our visual presentation result in Figure 8. We will also put the FID results in revision paper.
> > > >
> > > > **(2) Reply of Q1/Q2/Weakness 1 (about DG):** Thanks for your valuable suggestion, we will add these PACS experiments into the final version. We also will do our best to experiment with other datasets in DG for deeper exploration.
> > > >
> > > > **(3) Reply of Q3 and Q4:** Thanks for correcting our inappropriate expression. We have checked, and altered these inappropriate expressions in revision.
> > > >
> > > > ------
> > > >
> > > > Many thanks for your time! We hope that our response has addressed your remaining question.

---

> > > > ### Author Response · Authors · 2022-08-09
> > > > **Appreciate further comments**
> > > >
> > > > We would appreciate your above suggestions and comments.
> > > >
> > > > Please let us know whether we have addressed your concerns?
> > > >
> > > > Best regards,
> > > >
> > > > Authors of Paper 2664

---

### Official Review · Reviewer_haXr · 2022-07-08

**Rating:** 5
**Confidence:** 4
**Soundness:** 2 fair
**Presentation:** 3 good
**Contribution:** 2 fair

**Summary:**

The paper proposes a discrete adversarial training strategy to alleviate the robustness-generalization trade-off in vision tasks.  Motivated from the adversarial training used in NLP models, the authors claim that a discrete representation of the image space will aid in more robust models without much drop in generalization. The method utilizes a VQGAN model to generate discrete adversarial samples for adversarial training. The authors show that the adversarial training strategy enhances the performances of various vision tasks such as classification, object detection and self-supervised learning.

**Questions:**

- Please report robust accuracy values for epsilon = 4/255 on ImageNet and L_inf for AutoAttack.

- I understand that error bars on ImageNet are not possible - but for CIFAR, that would be easier - and allow for a comparison to more SotA methods on RobustBench. For CIFAR, the epsilon for the comparison should be 8/255 for L_inf on AutoAttack.





**Limitations:**

The authors already mention the limitations of the proposed method – the computational cost and lack of theoretical intuition. Although there are some empirical intuitions provided in the paper (fig 2), the missing error bars in all experiments make the empirical findings less convincing.

**Strengths And Weaknesses:**

+ Originality -  The idea to use discrete adversarial samples to enhance the performance of vision tasks seems novel/interesting.

+ Clarity -  The paper is well written and can be easily understood.

- Clarity -  I felt some of the experimental details are missing. For example, the implementation details of the adversarial  robustness experiments  - FGSM and Damagenet. There are some typos which needs to be proofread.

- Quality - The paper lacks theoretical intuition and the experimental section needs more stronger baselines for robustness tasks.  For example the intuition that the straight-through gradient computation and backpropagation canjust be skipped seems very odd to me. At least some empirical evidence (comparison of the adversarial examples that are actually computed by backpropagation) should be reported.

- Significance - It is difficult to analyze the superiority of the method in adversarial robustness since the robustness experiments is missing baselines from RobustBench. However, number for much smaller epsilons than the ones usually evaluated in robustBench indicate that the method strongly underperforms in this respect.
- Without error bars, it is unclear how significant the results are.


After the rebuttal and extentive discussion, I see that the proposed approach might have potential w.r.t. model generalization. My orignial argument w.r.t. adversarial robustness holds, but I am willing to increase my score based on the presented results.

---

> ### Author Response · Authors · 2022-08-02
> **Response to Reviewer haXr (1 of 2)**
>
> We thank the reviewer for the time and constructive comments.
>
> ------
> **Q1:** missing implementation details of the adversarial robustness experiments - FGSM and Damagenet. There are some typos which needs to be proofread.
>
> **Reply:** Sorry for the unclarity. For FGSM, we take sign operation on backward adversarial gradient, and multiply it with epsilon=1/255 to get the perturbation. We add the perturbation on original input to get adversarial examples, which is inferred by model to calculate top@1 accuracy. For DamageNet, it consists of 50000 adversarial examples which is pre-generated in [46]. [46] claims these images can fool ImageNet models to have error rate up to 85%, and can be used for evaluating the model performance against transferable adversarial attacks. In this work, we directly inference on these examples and calculate top@1 accuracy.
>
> We have carefully checked the typos and improved the writing in the revised manuscript.
>
> ------
>
> **Q2:** The paper lacks theoretical intuition and the experimental section needs more stronger baselines for robustness tasks. For example the intuition that the straight-through gradient computation and backpropagation can just be skipped seems very odd to me. At least some empirical evidence (comparison of the adversarial examples that are actually computed by backpropagation) should be reported.
>
> **Reply:** There are two places using straight-through gradient estimator. The first is in Line 159, it is widely used in VQVAE [36]. VQVAE adopts the technique proposed in [a] to learn the non-differentiable vector quantization module. [a] shows straight-through method has the right sign to “back-propagate” the non-smooth neurons. Another work [b] provided the theoretical justification for how the straight-throught estimator minimizing the training loss. Till now, straight-through gradient estimator is good performed in VQGAN optimization. So we followed this method. The citation of two papers ([a], [b]) has also been added in latest revision.
>
> The second straight-through gradient estimator from $Q(x)$ to $x$ is in Line 169. It is necessary for DAT, because the GPU memory cost will become 4× and training time will get 8× longer if we do not make this assumption. It is indeed a strict assumption. The reviewer may concern such hypothesis will fail in practice. Therefore, we show some empirically results to explore how this hypothesis behaves in not idea setting:
>
> **1) The similarity of the backward gradients on $Q(x)$ and $x$.** We add the visualization of backward gradients on $Q(x)$ and $x$ in Appendix D.6. It suggests the high visual similarity of the gradients on $Q(x)$ and $x$.
>
> **2) The attack ability of discrete adversarial examples under this assumption.** The reviewer may concern, in practice (not in ideal setting), if the straight-through method can still accurately estimate the direction of the adversarial  gradient. To relieve this concern, we present some results below, which suggest the discrete adversarial examples crafted by straight-through gradient is still with strong attack ability. It demonstrates the gradient of $Q(x)$ is effective for approximating the adversarial gradients on $x$.
> We compare the attack strength of discrete adversarial examples bellow:
>
> |  Type of AEs  | $\epsilon$ | Different VQGAN | Attack Suc. Rate  |
> |  ----  | ----  |  ----  |  ----  |
> | FGSM [45]  | 1/255 | - | 87.81% |
> | Discrete AE w/ Backward Gradients [36]  | - | VQGAN with FID=1.14 | 84.62% |
> | Discrete AE w/ Straight-Through Gradients  | - | VQGAN with FID=1.14 | 82.44% |
> | Discrete AE w/ Backward Gradients [36]  | - | VQGAN with FID=4.98 | 83.56% |
> | Discrete AE w/ Straight-Through Gradients  | - | VQGAN with FID=4.98 | 80.17% |
> | Discrete AE w/ Backward Gradients [36]  | - | VQGAN with FID=7.94 | 82.90% |
> | Discrete AE w/ Straight-Through Gradients  | - | VQGAN with FID=7.94 | 79.27% |
>
> We adopt pretrained resnet50 as target model. The attack success rate (87.81%) of FGSM [45] in pixel space is shown for reference. The better VQGAN model is used, the higher Attack Success Rate (ASR) can be achieved. Compared with FGSM, the AEs in discrete space with backward gradients [36] get slight drop on ASR caused by the information compression in discrete spaces. The straight-through method used in our DAT can produce AEs with 82.44%, 80.17% and 79.27% ASR for VQGAN with FID=1.14, 4.98, 7.94 respectively. It is few points lower than using the directly backward gradients, but can still keep relatively high attack strength, making nearly 80% examples misclassified.
>
> **Reference:**
>
> [a] “Estimating or Propagating Gradients Through Stochastic Neurons for Conditional Computation”, Yoshua Bengio, Nicholas Leonard and Aaron Courville
>
> [b] “Understanding Straight-Through Estimator in Training Activation Quantized Neural Nets”, Penghang Yin, Jiancheng Lyu, Shuai Zhang, Stanley Osher, Yingyong Qi, Jack Xin

---

> > ### Author Response · Authors · 2022-08-02
> > **Response to Reviewer haXr (2 of 2)**
> >
> > **Q3:** lack of adversarial robustness and comparison to more SotA methods on RobustBench; report robust accuracy values for epsilon = 4/255 on ImageNet and L_inf for AutoAttack; report robust accuracy values for the epsilon=8/255 on CIFAR10 and L_inf for AutoAttack
> >
> > **Reply:** It should be denoted that our DAT is not belonging to the field of adversarial robustness research [c,d,e]. The pure adversarial robustness [c,d,e] only cares about the performance under worst $l_{p}$ bounded perturbations, but sacrificing the clean performance. Instead, DAT focuses on general robustness, where many aspects need to be considered: clean performance, corruption robustness, generalization ability, adversarial robustness, transferability to downstream tasks, etc. Recently, a line of research [26, 27, 28, 32, 39, 53] is proposed in this general robustness field. Our DAT is the follow up work in this area. **All of them do not claim the SoTA results on $l_{p}$ bounded adversarial robustness, but aim to achieve comprehensive improvement on multiple types of robustness.**
> >
> > **Therefore, it is unreasonable to compare our method with SoTA adversarial robustness methods on RobustBench, since they actually belong to different research areas.** The standard protocol of AutoAttack (4/255 on ImageNet, 8/255 on CIFAR10) is proposed for evaluating methods in pure adversarial robustness area, which is also not appropriate for DAT evaluation. RobustBench actually has multiple leaderboards. We show on Leaderboard of ImageNet Common Corruptions (ImageNet-C) in RobustBench, our DAT can achieve the SoTA results with 73.61% robust accuracy. This result is also higher than many other methods not in the leaderboard, such as MAE[19], DrViT[39], PyramidAT[28], etc.
> >
> > However, the kind suggestions of the reviewer is appreciated. It is interesting to apply the idea of discrete adversarial training into the field of adversarial robustness research. We are happy to do some exploration along this direction.
> >
> > ------
> >
> > We hope that our response has addressed all of your concerns. Thank you for your time and feedback on our submission! Please don't hesitate to let us know if you have any remaining questions or concerns.
> >
> > **Reference:**
> >
> > [c] “Towards Deep Learning Models Resistant to Adversarial Attacks”, Aleksander Madry, Aleksandar Makelov, Ludwig Schmidt, Dimitris Tsipras, Adrian Vladu
> >
> > [d] "Theoretically Principled Trade-off between Robustness and Accuracy", Hongyang Zhang, Yaodong Yu, Jiantao Jiao, Eric P. Xing, Laurent El Ghaoui, Michael I. Jordan
> >
> > [e] “Adversarial Weight Perturbation Helps Robust Generalization”, Dongxian Wu, Shu-tao Xia, Yisen Wang

---

> ### Author Response · Authors · 2022-08-05
> **Whether our response has addressed your concerns?**
>
> Dear reviewer haXr,
>
> We did not receive any feedback on our response yet.
>
> Please can you let us know whether you've read our rebuttal and whether we addressed your concerns?
>
> If we did not, please let us know what we failed to address appropriately.
>
> Thanks!!
>
> Authors of Paper 2664

---

> > ### Comment · Reviewer_haXr · 2022-08-05
> > **Significance and adversarial robustness**
> >
> > Thank you for the reply, the clarification and the additional results on straight though gradients. To be honest, the results indicate that it would actually be beneficial to compute the gradients and that the crude approximation of the opimization without gradients bares disadantages. Still, these results are of course valuable, especially when asusming that the computation might be affordable in specific scenarios (e.g. very low resolution data).
> >
> > While this addresses some of my concerns, I am still not convined by the argumentation on the significance of results. Also, I am not convinced by the argumentation w.r.t. adversarial robustness. The proposed method is a particularly expensive adversarial training scheme - therefore it should also be possible to train an adversarially robust model, i.e. a model that shows at least some robustness in the standard setting employed in RobustBench.

---

> > > ### Author Response · Authors · 2022-08-06
> > > **More discussion about the concerns about adversarial robustness**
> > >
> > > Thank you for your comments of our reply. We are very happy that part of the concerns has been resolved, and many thanks to the reviewers for helping us improve the paper.
> > >
> > > In this reply we will discuss the concerns about adversarial robustness.
> > >
> > > **(1)** Different from that PGD attack is usually used in traditional AT [c,d,e], DAT generates discrete AEs which exceed Lp bound for training. We have discussed the difference between discrete AEs and PGD AEs in **Weakness3 of reviewer C7HU**. Under this AT scheme, DAT actually builds the adversarial robustness against discrete adversarial attacks beyond Lp bound. We conduct an experiment to validate our claims. We use unrestricted AEs in discrete space to attack some robust models on ImageNet from https://github.com/microsoft/robust-models-transfer. The results are shown below:
> > > | Models | Clean Accuracy | Robust Accuracy |
> > > |  ----  | ---- | ---- |
> > > | Vanilla R50 | 76.13% | 13.56% |
> > > | Linf_robust R50 with eps=4/255 [g] | 68.46%  | 60.12% |
> > > | L2_robust R50 with eps=3.0 [g] | 62.83%  | 53.20% |
> > > | DAT (Ours) | 76.52% | 72.89% |
> > >
> > > DAT achieves 72.89% robust accuracy which is 12.77% higher than 4/255-linf-robust model. **Therefore, even if linf-robust R50 gets top@1 rank on RobustBench, in this experiment we suggest linf-robust models are only shown certain robustness under specific settings (Linf bound adversarial attacks).** It is less robust than our DAT on unrestricted attacks which appear more commonly in the real world.
> > > Overall, we think a suitable adversarial robustness metric is decided by the configuration of AEs used for training. RobustBench adopts 4/255 AutoAttack for evaluation since all the compared methods bound the training AEs into (-4/255, 4/255). But our DAT is trained beyond lp norm bounding, so evaluating DAT by 4/255 AutoAttack is a biased comparison (same as evaluating linf robust models by unrestricted attacks).
> > >
> > > **(2)** Our DAT does have expensive adversarial training scheme, however adopting AT scheme is not always meaning to SoTA adversarial robustness. There are multiple previous significant works [26, 28, f] adopt expensive adversarial training scheme, but they do not train a SoTA linf-robust model in RobustBench, instead they use AT scheme to improve the generalization. Their results are also of great significance in the field of image classification. Similarly, we should denote that the goal of this work is also improving generalization, but not achieving Lp adversarial robustness.
> > >
> > > We hope that our response has addressed all of your concerns. Thank you for your time and feedback! Please don't hesitate to let us know if you have any remaining questions or concerns.
> > >
> > > ------
> > >
> > > **Reference**
> > >
> > > [f] Improving Vision Transformers by Revisiting High-frequency Components, Bai, Jiawang and Yuan, Li and Xia, Shu-Tao and Yan, Shuicheng and Li, Zhifeng and Liu, Wei; In European Conference on Computer Vision, 2022.
> > >
> > > [g] Salman, Hadi, et al. "Do adversarially robust imagenet models transfer better?." Advances in Neural Information Processing Systems 33 (2020): 3533-3545.

---

> > > > ### Comment · Reviewer_haXr · 2022-08-07
> > > > **even more discussion**
> > > >
> > > > Thank you for your reply.
> > > >
> > > > - If I understand the new numbers correctly, the results proposed in the new table show that indeed the proposed model is robust against the attacks it is trained on. This is a valuable evaluation. However, this result is not surprising. The other evaluation, the number for all these models on RobustBench, would be more interesting in my opinion, because it would further show the generalization of the proposed model beyond the training data.
> > > > - If I understand the method correctly, it is at least as expensive as pgd adversarial training (because pgd attack is the first step). It is even more expensive because of the forward pass through the AE (which might be fast).  Therefore, in my understanding, the resulting model should also be at least as robust as a model trained using pdg AT. Of course, the trade-off can be different, say, higher clean accuracy, slightly lower robust accuracy. I would still like to see the numbers to get an impression!
> > > > - [g] implies that adversarially robust models can generalize better. Therefore, the finding proposed in this work, that an adversarial training scheme can generalize better to e.g. common corruptions, is to be expected. If you can not show robust accuracy on RobustBench, I would like to see a quantification of the benefit of this more complicated training scheme with extra hyperparameters over (finetuned!) traditional (pgd) adversarial training to see a fair comparison.

---

> > > > > ### Author Response · Authors · 2022-08-09
> > > > > **A fair comparison of our DAT with traditional AT (1 of 2)**
> > > > >
> > > > > Thank you for your time and helpful suggestions. We answer your questions as follows.
> > > > >
> > > > > **(1) About the training cost**
> > > > >
> > > > > We agree with you that training time is indeed an important factor in model performance. As far as we know, in traditional PGD-based AT[5] and other variants like TRADES[k], the used PGD attacker is not only one step. From the implementation in GitHub [h, i], AT always use PGD with 10 attack steps for training. Both of them require multiple gradient backward, while our DAT only need once. We show the training budget below:
> > > > > |  Training Strategies  | Attack steps used for training | Training budget  |
> > > > > |  ----  | ----  |  ----  |
> > > > > | Normal  | 0 | 1× |
> > > > > |  AdvProp [26]  | 1 | 3× |
> > > > > |  Adversarial Training [5]  | 10 | 11× |
> > > > > |  DAT(Ours) | 1 | 3.5× |
> > > > >
> > > > > Based on the above table,  it needs to be clarified that our training cost is much less than the traditional pgd adversarial training. Traditional AT with 10 attack steps has nearly 3× training costs than ours.
> > > > >
> > > > > **(2) The study of the generalization on traditional AT models**
> > > > >
> > > > > Thanks for your suggestion about generalization on traditional AT models.  We first explore how a traditional AT scheme effects on the model generalization.  We collect some open-sourced robust models[j] using resnet50 as backbone on ImageNet and test them on OOD datasets. The training cost of each robust model is also counted. The results are shown in below table:
> > > > > | AT models | Training Cost | ImageNet-Val | A | C(mCE↓) | V2 | R | Sketch | Stylized |
> > > > > |  ----  | ---- | ---- | ---- | ---- | ---- | ---- | ---- | ---- |
> > > > > | Normal training, $\epsilon=0$ | 1× |76.13 | 0.0 | 76.70 | 63.20 | 36.17 | 24.09 | 7.38 |
> > > > > | L2-Robust, $\epsilon=0.01$ [g] | 4× | 75.68 | 2.11 | 75.33 | 64.00 | 35.98 | 23.55 | 7.47 |
> > > > > | L2-Robust, $\epsilon=0.03$ [g] | 4× | 75.76 | 2.17 | 75.36 | 63.66 | 36.18 | 23.98 | 8.18 |
> > > > > | L2-Robust, $\epsilon=0.05$ [g] | 4× | 75.59 | 2.19 | 75.65 | 63.37 | 36.48 | 23.90 | 8.51 |
> > > > > | L2-Robust, $\epsilon=0.1$ [g] | 4× | 74.78 | 2.13 | 75.42 | 62.64 | 36.90 | 23.85 | 9.18 |
> > > > > | L2-Robust, $\epsilon=0.25$ [g] | 4× | 74.14 | 2.28 | 75.79 | 62.20 | 37.57 | 24.33 | 10.07 |
> > > > > | L2-Robust, $\epsilon=0.5$ [g] | 4× | 73.16 | 2.19 | 75.91 | 60.48 | 38.03 | 23.49 | 10.99 |
> > > > > | L2-Robust, $\epsilon=1.0$ [g] | 4× | 70.43 | 2.19 | 78.36 | 57.36 | 38.21 | 22.63 | 11.07 |
> > > > > | L2-Robust, $\epsilon=3.0$ [g] | 4× | 62.83 | 1.97 | 83.84 | 49.45 | 36.48 | 20.40 | 10.48 |
> > > > > | L2-Robust, $\epsilon=5.0$ [g] | 4× | 56.13 | 1.71 | 88.98 | 43.04 | 32.75 | 16.82 | 9.13 |
> > > > > | Linf-Robust, $\epsilon=0.5/255$ [g] | 4× | 73.73 | 2.35 | 76.86 | 61.88 | 38.54 | 23.79 | 10.94 |
> > > > > | Linf-Robust, $\epsilon=1.0/255$ [g] | 4× | 72.05 | 2.53 | 78.34 | 59.60 | 40.13 | 23.70 | 12.10 |
> > > > > | Linf-Robust, $\epsilon=2.0/255$ [g] | 4× | 69.10 | 2.52 | 80.09 | 56.64 | 38.65 | 22.14 | **12.36** |
> > > > > | Linf-Robust, $\epsilon=4.0/255$ [g] | 4× | 63.86 | 2.25 | 85.14 | 51.39 | 38.25 | 20.94 | 11.70 |
> > > > > | Linf-Robust, $\epsilon=8.0/255$ [g] | 4× | 54.53 | 2.12 | 91.59 | 42.16 | 34.40 | 18.10 | 9.58 |
> > > > > | FreeAT, $\epsilon=4.0/255$ [m] | **1×** | 59.96 | 1.62 | 90.26 | 47.39 | 35.72 | 17.46 | 10.34 |
> > > > > | **DAT(Ours)** | 3.5× | **76.52** | **4.38** | **74.16** | **65.02** | **41.90** | **27.27** | 10.8 |
> > > > >
> > > > > From the experimental results, we can summarize the following points:
> > > > > - Compared to normal training,  adversarial training will hurt the clean performance in the imagenet-val dataset even with extreme small perturbations.
> > > > > - The results suggest AT with a very small $\epsilon$ can slightly benefit from the generalization, e.g., with L2-Robust, $\epsilon=0.01$ , ImageNet-C mCE value from 76.70 dropped to 75.33, lower mCE means better common corruption generalization. But with the \epsilon becoming larger, AT greatly damages the generalization, e.g. with L2-Robust, $\epsilon=5.0$, ImageNet-C mCE value increases to 88.98. This finding is also revealed by [l].
> > > > > - Compared with traditional AT models, our DAT can improve generalization more significantly. It even surpasses the normal training on clean performance of imagenet-val dataset.

---

> > > > > > ### Author Response · Authors · 2022-08-09
> > > > > > **A fair comparison of our DAT with traditional AT (2 of 2)**
> > > > > >
> > > > > > **(3) The fair comparison of DAT and traditional AT**
> > > > > >
> > > > > > Thanks for your suggestion about the fair comparison. Here, we provide a totally fair comparison between DAT and traditional Adversarial Training by keeping consistent on the hyperparameters of  AT and DAT. Specially, we delete the discretization step, such that our DAT can degenerate into the traditional PGD-based AT. Since our DAT adopts one attack step and directly uses normalized gradients multiplied by $\alpha=0.1$ to craft AEs (without sign operation), it actually can be regarded as L2 adversarial training with attack_step=1 and $\epsilon=0.1$. So, we can use these hyperparameters to train a L2-robust model, and compare it fairly with our DAT models. The results are shown below:
> > > > > > | Models | $\epsilon$ | Perturbation Type | Attack Steps | ImageNet-Val | L2 AutoAttack $\epsilon=0.1$  | L2 AutoAttack $\epsilon=0.5$ | L2 AutoAttack $\epsilon=3$ | A | C(mCE↓) | V2 | R | Sketch | Stylized |
> > > > > > |  ----  | ----  |  ----  | ----  |  ----  | ----  |  ----  | ----  |  ----  | ----  |  ----  | ----  |  ----  | ----  |
> > > > > > | Traditional PGD-based AT | 0.1 | L2 | 1 | 74.48  | 60.58 | 9.71 | 0 | 1.65 | 80.48 | 61.72 | 36.30 | 22.90 | 7.37 |
> > > > > > | DAT (Ours) | unrestricted | Discrete | 1 | 76.52 | 60.74 | 9.6 | 0 | 4.38 | 74.16 | 65.02 | 41.90 | 27.27 | 10.8 |
> > > > > >
> > > > > > All the above models are using resnet50 as the backbone. From the results, we can see a clear quantification of the benefit of our proposed discrete AT scheme compared with traditional AT.
> > > > > >
> > > > > > **On clean performance:** traditional AT plays negative impact on ImageNet-val clean performance. However DAT can reduce the negative impact and achieve 2 points higher accuracy on ImageNet-val. It even surpasses the normal training on clean performance of imagenet-val dataset.
> > > > > >
> > > > > > **On adversarial robustness:** we should admit that DAT indeed cannot yield significantly better adversarial robustness compared with traditional AT. But by presenting the L2 AutoAttack evaluation with different $\epsilon$, we show DAT at least can achieve a comparable adversarial robustness with traditional AT. We also add L2 AutoAttack evaluation with $\epsilon=3$, in this condition, both traditional AT and our DAT drop to 0% accuracy. It is expected because only $\epsilon=0.1$ is used in training.
> > > > > >
> > > > > > **On generalization:** the result shows DAT achieves significant improvement on generalization compared with traditional AT. It is the main contribution of this work, that is DAT can enhance the quality of learned representation. We also give some insights to explain why discrete representation can help generalization in Q1 of Reviewer oF1f. You can also refer to it for more details.
> > > > > >
> > > > > > For experiments with larger perturbations and more attack steps, sorry for that we do not compare them here because the training time is too long. We will add these comparisons in the final revision.
> > > > > >
> > > > > > Lastly, we hope we have clarified some of our procedures in the adversarial training and allayed some of your concerns. Moreover, we hope that the reviewer appreciates the value of discrete adversarial training  in ideas and methods  for boosting the performance on representation learning. Thank you for the comments. Please don't hesitate to let us know if you have any remaining questions or concerns.
> > > > > >
> > > > > > ------
> > > > > >
> > > > > > **Reference**
> > > > > >
> > > > > > [h] https://github.com/MadryLab/cifar10_challenge/blob/f15682d9f1e26eb47a2d3b371ef8b6c7abcf6276/config.json#L27
> > > > > >
> > > > > > [i] https://github.com/yaodongyu/TRADES/blob/6e8e11b7c281371c2f027ffadfbaea80361f09de/train_trades_cifar10.py#L32
> > > > > >
> > > > > > [j] https://github.com/microsoft/robust-models-transfer
> > > > > >
> > > > > > [k] Zhang, Hongyang, et al. "Theoretically principled trade-off between robustness and accuracy." International conference on machine learning. PMLR, 2019.
> > > > > >
> > > > > > [l] Kireev, Klim, Maksym Andriushchenko, and Nicolas Flammarion. "On the effectiveness of adversarial training against common corruptions." UAI (2022).
> > > > > >
> > > > > > [m] Shafahi, Ali, et al. "Adversarial training for free!." Advances in Neural Information Processing Systems 32 (2019).

---

> > > > > > > ### Comment · Reviewer_haXr · 2022-08-09
> > > > > > > **additional results and fair comparison**
> > > > > > >
> > > > > > > Thank you for the evaluation. In terms of fair comparison, I meant that the hyperparameters need to be finetuned for both settings (assuming that they are finetuned for DAT already). In this respect, the proposed evaluation is NOT fair. I understand that a fair comparison is likely not possible given the short time of the discussion phase .
> > > > > > > How are the hyperparameters selected for DAT  (grid search, BO, trial&error)?

---

> > > > > > > > ### Author Response · Authors · 2022-08-09
> > > > > > > > **Reply of the training hyperparameters**
> > > > > > > >
> > > > > > > > We do not use any hyperparameters search algorithms, since ImageNet training is very costly. In our implementation, we just use the default parameters for imagenet training[22], the compared traditional AT baseline is using the same hyperparameters too. For clarity, we list the training hyperparameters here:
> > > > > > > >
> > > > > > > > **basic hyperparameters:**
> > > > > > > >
> > > > > > > > epochs: 90
> > > > > > > >
> > > > > > > > batch_size: 128
> > > > > > > >
> > > > > > > > optimizer: sgd
> > > > > > > >
> > > > > > > > lr: 0.1
> > > > > > > >
> > > > > > > > lr_schedule: step
> > > > > > > >
> > > > > > > > lr_decay_epochs: 30
> > > > > > > >
> > > > > > > > lr_decay_rate: 0.1
> > > > > > > >
> > > > > > > > weight_decay: 1e-4
> > > > > > > >
> > > > > > > > **attacker hyperparameters (DAT without discrete procedures is the same with L2 based PGD):**
> > > > > > > >
> > > > > > > > perturbation_type: L2
> > > > > > > >
> > > > > > > > epsilon: 0.1 ( \alpha: 0.1 in DAT)
> > > > > > > >
> > > > > > > > attack_step: 1
> > > > > > > >
> > > > > > > > attack_step_size: 0.1
> > > > > > > >
> > > > > > > > **used data augmentation:**
> > > > > > > >
> > > > > > > > data_augmentation: https://github.com/MadryLab/robustness/blob/a9541241defd9972e9334bfcdb804f6aefe24dc7/robustness/data_augmentation.py#L40
> > > > > > > >
> > > > > > > > The training is conducted on single machine with 8 GPU cards. We will also open source the training code, for the reproducibility of our work. Hope the above will solve your concerns. Thanks for your time and comments.

---

> > > > > > > > > ### Comment · Reviewer_haXr · 2022-08-09
> > > > > > > > > **hyperparameters**
> > > > > > > > >
> > > > > > > > > okay, thank you for the clarification

---

> > > > > > > ### Comment · Reviewer_haXr · 2022-08-09
> > > > > > > **thank you for the evaluation**
> > > > > > >
> > > > > > > agreed - thank you for the evaluation of adversarial robustness.

---

> > > > > > > > ### Author Response · Authors · 2022-08-09
> > > > > > > > **Thanks for your affirmation**
> > > > > > > >
> > > > > > > > Dear reviewer haXr,
> > > > > > > >
> > > > > > > > We are happy to get affirmation from you about the experiment results. We will add these experiments in our revision. Many thanks again for your precious review time and valuable comments to help us improve the paper.
> > > > > > > >
> > > > > > > > Best,
> > > > > > > >
> > > > > > > > Authors of Paper 2664

---

> ### Author Response · Authors · 2022-08-09
> **Are there any remaining unresolved concerns**
>
> Dear reviewer haXr,
>
> We are encouraged and thankful for your rating improving.  However, the rating is still negative. We think that we've addressed your concerns through detailed experiments including  generalization of adversarial trained models and fair comparison between DAT and PGD adversarial Training. The experimental results also suggest that our method has comparable adversarial robustness.
>
> If there are remaining unresolved concerns, we are happy to continue the discussion.
>
> Thanks!!
>
> Authors of Paper 2664

---

### Official Review · Reviewer_oF1f · 2022-07-11

**Rating:** 6
**Confidence:** 3
**Soundness:** 3 good
**Presentation:** 4 excellent
**Contribution:** 3 good

**Summary:**

This paper proposes discrete adversarial training to boost the performance on representation learning. Extensive results on image classification, object detection and self-supervised learning validate the effectiveness.

**Questions:**

See above.

**Strengths And Weaknesses:**

## Strengths:

The presentation is clear and extensive experimental results are convincing.

## Weaknesses:

1. My biggest concern is why discrete representation helps?

2. DrVit is very similar except for inner optimization and quantization training. DrVit is training VQVAE from scratch. Will that become unfair comparison? For example, you are using different data for training or pretraining. Could you provide the results if the proposed method is also trained from scratch using the same dataset, loss and architecture? Or DrVit is also using VQGAN

3. In Table 1, why the winner between DrVit and AugReg-Vit keeps changing on different metrics?

---

> ### Author Response · Authors · 2022-08-02
> **Response to Reviewer oF1f (1 of 2)**
>
> We thank the reviewer for the time and insightful comments.
>
> **Q1:** why discrete representation helps
>
> **Reply:** It is an interesting problem and worth deeply studying. Since there are few works exploring the discrete representation in AT before, why discrete representation helps AT is still an open question.
>
> We provide our insights about the help of discrete representation in DAT, which lies in two aspect:
>
> **(1)** As shown in [36], instead of focusing or spending capacity on image pixel level noise and imperceptible local details, discrete representation captures important features, preserves the global structure and semantics of an object. So crafting AEs in discrete symbolic space yields more meaningful semantic perturbations for our DAT training. To provide some evidence for the more meaningful semantic perturbations on discrete representation, we add more discussion and comparison of discrete AEs and pixel-space AEs in Appendix D.5. For fairness, we keep the same attack success rate of all compared AEs. The study has three aspects:
>
> - *Discrete perturbations create more realistic AEs.* We add a visualization of pixel-wise AEs and discrete AEs in Figure 8 of Appendix D.5 for subjective photorealism comparison. Pixel-wise perturbations lead to noisy images. By calculating the number of colors [17], we find pixel-wise AEs add more  invalid colors, resulting in a noisy image. While discrete perturbations have minor changes on the color numbers of original image. Such subtle change is hard to be perceived by humans.
> - *Discrete perturbations have more low frequency component.* We conduct frequency analysis on compared AEs in second row of Figure 8. Pixel-wise perturbations introduce more high frequency component. It may lead the pixel-wise AEs to far away from natural distributions. However discrete perturbations will not introduce unnecessary high-frequency components in original image.
> - *Discrete perturbations are more structural.* From the perturbation visualization in third row of Figure 8 , we find discrete perturbations have more structured information about objects, shown it attends to more important locations. While pixel-wise perturbations are noisy and disordered.
>
> **(2)** From the perspective of distribution, we show in Line 193 that the AEs crafted based on discrete representation are closer to the natural image distribution. It can reduce the underlying distribution shift caused by pixel-space AEs [26], and enhance the robustness and generalization without sacrificing clean performance.
>
> Above we show some superior properties of discrete AEs. Several of properties are shown benefit for AT in previous works [26,28,a]. [28] shows the structured adversarial perturbations can achieve significant performance gains over non-adversarial baseline and adversarial training with pixel perturbations. [26] shows solving the distribution shift problem of clean and adversarial examples in AT can help the clean performance and generalization. [a] thinks the latent space contains compressed semantic-level features. So AT on the perturbations generated in latent space may guide the classification model to use robust features instead of achieving high accuracy by exploiting non-robust features in the image space.
>
> Overall, we think such superior properties of discrete AEs contribute to the good performance of our DAT. We are also welcome any other insights in the future study to discuss this open problem.
>
> **Reference:**
>
> [a] “Dual Manifold Adversarial Robustness: Defense against Lp and non-Lp Adversarial Attacks”, Wei-An Lin, Chun Pong Lau, Alexander Levine, Rama Chellappa, Soheil Feizi

---

> > ### Author Response · Authors · 2022-08-02
> > **Response to Reviewer oF1f (2 of 2)**
> >
> > **Q2:** unfair comparison to DrVit
> >
> > **Reply:** The biggest difference with DrViT is that DrViT only discretizes the input for training, while our DAT conducts an adversarial process in discrete space to generate more diverse and harder discrete adversarial examples for training. So DAT regularizes the model learning more robust and generalized representation.
> >
> > We then give a completely fair experimental comparison. After checking the official implementation, we find DrViT actually uses VQ-GAN model with $k=1024$, $d=256$ for discretization. Instead of training both VQ-GAN and ViT classification model from scratch, DrViT consists of two stage training: 1) it first pretrains the VQ-GAN encoder and decoder on ImageNet; 2) then it finetunes the discrete embeddings learned in first stage.
> > To provide more fair comparison, we modify our DAT to use the VQ-GAN model with $k=1024$, $d=256$ pretrained on ImageNet. By aligning all settings on dataset and architecture, a completely fair comparison to DrViT is below:
> >
> > | Models | ImageNet-Val | FGSM | DamageNet | A | C↓ | V2 | R | Sketch | Stylized |
> > |  ----  | ---- | ---- | ---- | ---- | ---- | ---- | ---- | ---- | ---- |
> > | DrViT  | 79.48 | 45.76 | 44.91 | 17.20 | 46.22 | 68.05 | 44.77 | 34.59 | 19.3 |
> > | DAT (Ours) | 80.46 | 51.17 | 49.49 | 24.75 | 45.14 | 68.71 | 47.51 | 35.64 | 22.96 |
> >
> > Under this fair comparison, our DAT still achieves better performance on both robustness and generalization.
> >
> > ------
> >
> > **Q3:** why the winner between DrVit and AugReg-Vit keeps changing on different metrics?
> >
> > **Reply:** We admit that DrViT and AugReg-ViT exactly do inconsistent effect on different robustness metrics in Table 1. After careful check, we confirm all the results are accurate and convincing. Such a phenomenon is expected, as DrViT strengthens the robustness by improving the ability of capturing shape features, and this behaviour is just beneficial for some robustness metrics. For example, learning shape feature shows effectiveness on ImageNet-C, -R, -Sketch, while it is not useful for ImageNet-A, which contains hard nature samples. Other inconsistence on DamageNet is rational, as the gap is marginal, showing both DrViT and AugReg-ViT perform similarly on DamageNet.
> >
> > ------
> >
> > We hope that our response has addressed all of your concerns. Thank you for your time and feedback on our submission! Please don't hesitate to let us know if you have any remaining questions or concerns.

---

### Official Review · Reviewer_wZbH · 2022-07-11

**Rating:** 7
**Confidence:** 4
**Soundness:** 2 fair
**Presentation:** 3 good
**Contribution:** 3 good

**Summary:**

The paper proposes Discrete Adversarial Training (DAT) strategy for vision tasks that avoids continuous-Lp based perturbations and encourages adversarial examples generated through image discretization using VQGAN. Authors derive the motivation for using image discretization from the observation of improved generalization via discrete adversarial training in NLP tasks. Adversarial examples generated using this discretization process are used for training different architectures for different downstream tasks. Unlike traditional adversarial training (AT), DAT shown to improve network generalization across different tasks, different distribution shifts and also improve adversarial robustness. Major improvements are noticed in image classification task and minimal or comparable performance seen on object detection and semantic segmentation tasks.

**Questions:**

In line 148, it is mentioned that “We delete the constraint term since there is no need to bound the per-pixel values of \delta”. Would you elaborate on why this constraint is not needed?

In line 157, authors mention “as proposed in previous work, a straight-through gradient estimator can be used by copying the gradients from v_q to v”. Which previous work is referred here? Would you provide justification how the gradients can be estimated and copied from v_q to v?

Similarly a straight-through gradient estimator was used between Q(x) and x. How does this hold in practice (not in ideal setting) ?

In line 179, it is mentioned that  “x + \delta is discretized by VQGAN again and acts as the adversarial input”. Do authors cross check that these adversarial inputs generated by VQGAN still mislead the classifier F?


**Limitations:**

Authors have not discussed about their limitations. I find no potential negative impact from this work. The method requires heavily parameterized VQGAN to train different networks. This imposes heavy computation time and budget during the training process.

**Strengths And Weaknesses:**

 Strengths:
1)	Well written paper. Easy to understand.
2)	Motivation is explained clearly.
3)	Major strength lies in the extensive experimental evaluation.
4)	Shown improvements of network generalization across different ImageNet distribution shift datasets and higher adversarial robustness on both ResNet50 and ViT.
5)	Conducted ablation studies to understand the effect of different components.

Weakness:
Different assumptions are made to ease the computation of gradients, backpropagation through VQGAN to generate adversarial examples. No theoretical details are shown to support the assumptions and approximations are made through heuristics.

---

> ### Author Response · Authors · 2022-08-02
> **Response to Reviewer wZbH (1 of 2)**
>
> We thank the reviewer for the time and constructive comments.
>
> **Q1:** “We delete the constraint term since there is no need to bound the per-pixel values of $\delta$” why this constraint is not needed？
>
> **Reply:** The reason of “no need to bound the per-pixel values of \delta” lies in three aspect:
>
> **(1)** DAT bounds the final perturbation by first effecting on pixel space and further impact the discrete space. Therefore $\delta$ is just an intermediate result in the process of computing the final symbolic perturbations. To study the effect of per-pixel bound on $\delta$, we first use $\alpha=0.1$ to generate 1000 samples and count the proportion of the $\delta$ in different perturbation intervals:
> | Intervals  | Proportion of the $\delta$ in the interval |
> |  ----  | ----  |
> | (-1/255, 1/255) | 28.3% |
> | (-2/255, 2/255) | 95.9% |
> | (-4/255, 4/255) | 100% |
>
> It shows 95.9% of the $\delta$ is in (-2/255, 2/255), and all $\delta$ are in (-4/255, 4/255). Using magnitude $\alpha$ has almostly regulated the $\delta$ into (-4/255, 4/255). So adding the other per-pixel bound on $\delta$ seems unnecessary.
>
> **(2)** It is unclear if adding per-pixel bound on \delta will impact the performance. To study this problem, we bound the $\delta$ with different epsilon and re-run the DAT.
> | Linf Bounds on $\delta$ | ImageNet-Val | FGSM | DamageNet | A | C↓ | V2 | R | Sketch | Stylized |
> |  ----  | ---- | ---- | ---- | ---- | ---- | ---- | ---- | ---- | ---- |
> | no bounds  | 76.52 | 30.66 | 14.42 | 4.38 | 74.16 | 65.02 | 41.90 | 27.27 | 10.8 |
> | (-4/255, 4/255) | 76.47 | 31.43 | 14.25 | 4.31 | 74.12 | 65.07 | 41.68 | 26.99 | 10.62 |
> | (-2/255, 2/255) | 76.16 | 29.75 | 13.24 | 3.75 | 74.87 | 64.32 | 40.38 | 25.53 |9.31 |
> | (-1/255, 1/255)| 76.10 | 29.41 | 12.00 | 3.53 | 75.53 | 64.11 | 39.05 | 25.04 | 8.69 |
>
> As shown in above table, DAT achieves best performance when $\delta$ is not bounded. The worst result is appeared when $\delta$ is bounded between (-1/255, 1/255). With larger $l_{\infty}$ bound, the results become better. This experiment is added into Appendix B.6 of the revised paper.
>
> **(3)** As stated in Line 59, a good property of DAT is that it can produce diverse adversarial inputs beyond Lp bound for training. Bounding the per-pixel values of $\delta$ may potentially reduce the diversity of the discrete adversarial examples for training in our DAT.

---

> > ### Author Response · Authors · 2022-08-02
> > **Response to Reviewer wZbH (2 of 2)**
> >
> > **Q2:** citation and justification of straight-through gradient estimator
> >
> > **Reply:** There are two places using straight-through gradient estimator. The first is in Line 159, it is widely used in VQVAE [36]. VQVAE adopts the technique proposed in [a] to learn the non-differentiable vector quantization module. [a] shows straight-through method has the right sign to “back-propagate” the non-smooth neurons. Another work [b] provided the theoretical justification for how the straight-throught estimator minimizing the training loss. Till now, straight-through gradient estimator is good performed in VQGAN optimization. So we followed this method. The citation of two papers ([a], [b]) has also been added in latest revision.
> >
> > The second straight-through gradient estimator from $Q(x)$ to $x$ is in Line 169. It is necessary for DAT, because the GPU memory cost will become 4× and training time will get 8× longer if we do not make this assumption. It is indeed a strict assumption. The reviewer may concern such hypothesis will fail in practice. Therefore, we show some empirically results to explore how this hypothesis behaves in not idea setting:
> >
> > **(1)** The similarity of the backward gradients on $Q(x)$ and $x$. We add the visualization of backward gradients on $Q(x)$ and $x$ in Appendix D.6. It suggests the high visual similarity of the gradients on $Q(x)$ and $x$.
> >
> > **(2)** The attack ability of discrete adversarial examples under this assumption. The reviewer may concern, in practice (not in ideal setting), if the straight-through method can still accurately estimate the direction of the adversarial  gradient. To relieve this concern, we present some results in Q3, which suggest the discrete adversarial examples crafted by straight-through gradient is still with strong attack ability. It demonstrates the gradient of $Q(x)$ is effective for approximating the adversarial gradients on $x$.
> >
> > ------
> >
> > **Q3:** check that these adversarial inputs generated by VQGAN still mislead the classifier F
> >
> > **Reply:** We compare the attack strength of discrete adversarial examples bellow:
> > |  Type of AEs  | $\epsilon$ | Different VQGAN | Attack Suc. Rate  |
> > |  ----  | ----  |  ----  |  ----  |
> > | FGSM [45]  | 1/255 | - | 87.81% |
> > | Discrete AE w/ Backward Gradients [36]  | - | VQGAN with FID=1.14 | 84.62% |
> > | Discrete AE w/ Straight-Through Gradients  | - | VQGAN with FID=1.14 | 82.44% |
> > | Discrete AE w/ Backward Gradients [36]  | - | VQGAN with FID=4.98 | 83.56% |
> > | Discrete AE w/ Straight-Through Gradients  | - | VQGAN with FID=4.98 | 80.17% |
> > | Discrete AE w/ Backward Gradients [36]  | - | VQGAN with FID=7.94 | 82.90% |
> > | Discrete AE w/ Straight-Through Gradients  | - | VQGAN with FID=7.94 | 79.27% |
> >
> > We adopt pretrained resnet50 as target model. The attack success rate (87.81%) of FGSM [45] in pixel space is shown for reference. The better VQGAN model is used, the higher Attack Success Rate (ASR) can be achieved. Compared with FGSM, the AEs in discrete space with backward gradients [36] get slight drop on ASR caused by the information compression in discrete spaces. The straight-through method used in our DAT can produce AEs with 82.44%, 80.17% and 79.27% ASR for VQGAN with FID=1.14, 4.98, 7.94 respectively. It is few points lower than using the directly backward gradients, but can still keep relatively high attack strength, making nearly 80% examples misclassified.
> >
> > ------
> >
> > We hope that our response has addressed all of your concerns. Thank you for your time and feedback on our submission! Please don't hesitate to let us know if you have any remaining questions or concerns.
> >
> > **Reference:**
> >
> > [a] “Estimating or Propagating Gradients Through Stochastic Neurons for Conditional Computation”, Yoshua Bengio, Nicholas Leonard and Aaron Courville
> >
> > [b] “Understanding Straight-Through Estimator in Training Activation Quantized Neural Nets”, Penghang Yin, Jiancheng Lyu, Shuai Zhang, Stanley Osher, Yingyong Qi, Jack Xin

---

> > > ### Comment · Reviewer_wZbH · 2022-08-10
> > > **Thanks for the response**
> > >
> > > Authors response has convincingly addressed my concerns and I am willing to increase the score.

---

> > > > ### Author Response · Authors · 2022-08-10
> > > > **Appreciate for your precious review time and valuable comments**
> > > >
> > > > Dear reviewer wZbH,
> > > >
> > > > We are appreciate for getting an affirmation from you about our response. Many thanks again for your precious review time and valuable comments to help us improve the paper.
> > > >
> > > > Best,
> > > >
> > > > Authors of Paper 2664

---

### Author Response · Authors · 2022-08-02
**Update Manuscript**

We would like to thank all of the reviewers again for helping us improving our paper. We uploaded a revised version of our paper and marked the major modifications in blue for visibility. In short,

- We have carefully checked the typos and improved the writing in the revised manuscript.
- We have cited and discussed the papers the reviewers provided.
- We discussed the VILLA and AGAT in the Section.2 and declared the difference.
- We add discussion of limitation in Section.5
- We add a discussion of the necessity of bounding the per-pixel values of \delta in Appendix B.6
- We add a comparison of discrete perturbations with traditional pixel-space perturbations in Appendix D.5
- We add visualization of the straight-through gradients and directly backward gradients in Appendix D.6

Thank you all again for your precious and insightful suggestions. Please let us know if you have additional questions or ideas for improvement.

---

### Meta-Review · Area_Chair_Fc5J · 2022-08-28

**Recommendation:** Accept
**Confidence:** Certain

**Metareview:**

This paper proposes a discrete adversarial training scheme for improving the robustness of vision models. Reviewers find the paper is well written, the proposed idea seems to be novel/interesting, and the approach leads to improved empirical performance. This work may also inspire new approaches for improving both robustness as well as generalization together. Therefore, I recommend accepting the paper, while also encourage the authors to address the remaining issues pointed out by the reviewers.


**Award:**

No

---

### Decision · Program_Chairs · 2022-09-14

Accept